# PD-L1 positive astrocytes attenuate inflammatory functions of PD-1 positive microglia in models of autoimmune neuroinflammation

Mathias Linnerbauer[1,2,9], Tobias Beyer[2,9], Lucy Nirschl[2], Daniel Farrenkopf[1], Lena Lößlein [1], Oliver Vandrey[1], Anne Peter[1], Thanos Tsaktanis[1,2], Hania Kebir[3], David Laplaud [4], Rupert Oellinger [5,6], Thomas Engleitner[5,6], Jorge Ivan Alvarez[3], Roland Rad [5,6], Thomas Korn [2], Bernhard Hemmer [2], Francisco J. Quintana[7,8] & Veit Rothhammer [1,2] ✉

Multiple Sclerosis (MS) is a chronic autoimmune inflammatory disorder of the central nervous system (CNS). Current therapies mainly target inflammatory processes during acute stages, but effective treatments for progressive MS are limited. In this context, astrocytes have gained increasing attention as they have the capacity to drive, but also suppress tissue-degeneration. Here we show that astrocytes upregulate the immunomodulatory checkpoint molecule PD-L1 during acute autoimmune CNS inflammation in response to aryl hydrocarbon receptor and interferon signaling. Using CRISPR-Cas9 genetic perturbation in combination with small-molecule and antibody-mediated inhibition of PD-L1 and PD-1 both in vivo and in vitro, we demonstrate that astrocytic PD-L1 and its interaction with microglial PD-1 is required for the attenuation of autoimmune CNS inflammation in acute and progressive stages in a mouse model of MS. Our findings suggest the glial PD-L1/PD-1 axis as a potential therapeutic target for both acute and progressive MS stages.

Astrocytes are key players in the central nervous system (CNS) with versatile functions in health and disease[1,2]. In addition to their role in development and homeostasis including metabolic support for neurons[3,4] and control of blood-brain barrier (BBB) properties[5], astrocytes have the ability to react to inflammatory insults and to promote or inhibit CNS pathology depending on the surrounding micromilieu[6–8]. With the advent of single-cell methodologies, it has become clear that this functional diversity is associated to heterogenous activation states that depend on a plethora of environmental, cellular, or microbiome-derived cues[6,9–13]. While numerous studies have demonstrated that reactive astrocytes can actively drive pathogenesis and the propagation of neuro-inflammatory events[6,11,12,14], less is known about their regulatory and tissue-protective functions in the context of disease[13].

[1]Department of Neurology, University Hospital, Friedrich-Alexander University Erlangen Nuremberg, Erlangen, Germany. [2]Department of Neurology, Klinikum rechts der Isar, Technical University of Munich, Munich, Germany. [3]Department of Pathobiology, School of Veterinary Medicine, University of Pennsylvania, Philadelphia, PA, USA. [4]Nantes Université, INSERM, CNRS, Center for Research in Transplantation et Translational Immunology, UMR 1064, Nantes, France. [5]Institute of Molecular Oncology and Functional Genomics, Center for Translational Cancer Research (TranslaTUM), Technical University of Munich, Munich, Germany. [6]Department of Medicine II, Klinikum rechts der Isar, Technical University of Munich, Munich, Germany. [7]Ann Romney Center for Neurologic Diseases, Brigham and Women's Hospital, Harvard Medical School, Boston, MA, USA. [8]The Broad Institute of Harvard and MIT, Cambridge, MA, USA. [9]These authors contributed equally: Mathias Linnerbauer, Tobias Beyer. ✉e-mail: veit.rothhammer@fau.de

Based on their abundance and strategic localization in the CNS, astrocytes interact with a variety of CNS resident but also infiltrating cell types during autoimmune CNS inflammation[15]. This interaction is often bidirectional and involves the tight regulation of inflammatory signaling cascades through immunological checkpoints[11,16]. One of these checkpoints is represented by the interaction between programmed cell death 1 ligand 1 (PD-L1; also known as Cluster of differentiation 274; CD274), a member of the B7 family of ligands, and its cognate receptor PD-1 (also known as Programmed cell death protein 1; PDCD1). PD-L1 is predominantly expressed by antigen-presenting cells (APCs), while PD-1 can be expressed by T cells, in which activation of the PD-L1/PD-1 signaling cascade modulates processes such as proliferation, pro-inflammatory cytokine production, anergy, and apoptosis[17–19].

While this pathway has been particularly well investigated in the field of neoplastic diseases, it has recently been reported that also astrocytes have the capacity to express PD-L1 not only in the context of glioblastoma, but also traumatic brain injury, aging, and Alzheimer's disease[20–23]. This becomes particularly interesting in the context of autoimmune CNS diseases like Multiple Sclerosis (MS), where auto-reactive peripheral immune cells transgress into the CNS to drive a multifactorial pathology characterized by demyelination, axonal degeneration and gliosis[24,25].

The ability of reactive astrocytes to express PD-L1 in the context of autoimmune CNS inflammation may offer the potential to modulate the pathogenic properties of infiltrating, but also CNS-resident cells via cell contact dependent, and independent mechanisms. This may be of particular relevance in progressive stages of MS, where effective therapeutic strategies are limited. These stages are characterized by inflammatory processes and chronic inflammation-driven neurode-generation "trapped" behind a closed BBB[26]. In these lines, recent research on progressive MS has suggested that potential therapeutics for progressive stages of MS require a combination of anti-inflammatory and neuroprotective strategies, limiting both inflammatory and degenerative aspects of progressive MS[26,27].

Here, we investigate the spatiotemporal expression of PD-L1 by astrocytes in the context of acute and chronic autoimmune neuroinflammation. We identify type I and II interferons, as well as aryl hydrocarbon receptor (AhR) signaling as regulators of PD-L1 in astrocytes and demonstrate the relevance of astrocytic PD-L1 for acute CNS inflammation using CRISPR-Cas9 genetic perturbation models. Finally, we demonstrate that activated microglia respond to membrane-bound, and soluble PD-L1 derived from astrocytes, which limits their pro-inflammatory properties. Overall, therapeutic modulation of astrocytic PD-L1 signaling may offer the potential to attenuate the inflammatory capacity of infiltrating and CNS-resident cells in both acute and progressive stages of autoimmune CNS inflammation.

## Results

### Reactive astrocytes upregulate PD-L1 in response to acute autoimmune CNS inflammation

In order to investigate the spatiotemporal expression of PD-L1 by astrocytes in the context of autoimmune CNS inflammation, we induced experimental autoimmune encephalomyelitis (EAE) in wild-type (WT) C57Bl/6 mice by immunization with myelin oligodendrocyte glycoprotein 35-55 (MOG$_{35–55}$) in complete Freund's adjuvant (CFA) followed by pertussis toxin (PTx) injection. Next, we quantified the expression of astrocytic PD-L1 over the course of EAE by flow cytometry. At peak of disease, when peripheral immune cells infiltrate the CNS, brain and spinal cord astrocytes upregulated PD-L1 expression (Fig. 1a–c). Notably, while aside from astrocytes also microglia and infiltrating monocytes expressed PD-L1 during peak of EAE, the absolute number of PD-L1$^+$ astrocytes exceeded both PD-L1$^+$ microglia and monocytes (Supplementary Fig. 1a). Similarly, immunofluorescence staining of brain tissue obtained from inflammatory lesions in MS

patients revealed high expression of PD-L1 by GFAP$^+$ astrocytes in active white matter lesions of MS patients, while to a lesser extent also IBA1$^+$ microglia expressed PD-L1 (Fig. 1d). Of note, we observed almost no PD-L1 expression by astrocytes in normal appearing white matter (NAWM), suggesting that PD-L1 expression in astrocytes depends on their inflammatory activation (Supplementary Fig. 1b).

Next, we aimed to understand whether PD-L1 expression was associated with a pro-inflammatory or tissue-protective phenotype in astrocytes. To that end, we isolated PD-L1$^+$ and PD-L1$^-$ astrocytes by fluorescent activated cell sorting (FACS) at peak of EAE (Fig. 1e and Supplementary Fig. 1c) and assessed their transcriptional signature. In comparison to PD-L1$^-$ astrocytes, PD-L1$^+$ astrocytes showed reduced expression of pro-inflammatory genes (Ccl3, Ccl5, Nfkb1, Il6, Il1b, Ifng, Nos2, Tnf, Cd44), while tissue-protective genes (Tgfb1, Lif, Ptn, Ngf, Gdnf, Vegfa, Vegfb) were increased (Fig. 1f and Supplementary Fig. 1d). However, this phenotype was not exclusive, as PD-L1$^+$ astrocytes expressed higher levels of the activation associated genes Gfap, and Ccl2, while they downregulated Bdnf compared to PD-L1$^-$ astrocytes, collectively suggesting that PD-L1 expression by astrocytes is dependent on their activation.

This was recapitulated in a publicly available single-cell RNA-seq and spatial transcriptome dataset[9], demonstrating high expression of Cd274 by activated, interferon-responsive super-responder astrocytes that were closely associated with vessels, ventricles and brain surfaces, representing points of entry for infiltrating cells (Supplementary Fig. 1e).

Membrane-bound PD-L1 is cleaved actively by the metalloproteases ADAM10, ADAM17, MMP9 and MMP13, generating soluble PD-L1 (sPD-L1), which is able to activate membrane-bound PD-1[28]. While this mechanism has been demonstrated as a mechanism of immune evasion in neoplastic diseases[29], its relevance during autoimmune inflammation is unclear. To determine the regulation of sPD-L1 in the CNS compartment in the context of MS, we quantified the abundance of sPD-L1 in the cerebrospinal fluid (CSF) of patients in acute and progressive stages of MS. Consistent with the increase of astrocytic PD-L1 and the upregulation of metalloproteases that cleave PD-L1 into its soluble form during EAE (Supplementary Fig. 1f), levels of sPD-L1 were significantly increased in the CSF of patients with clinically isolated syndrome (CIS) and relapsing-remitting MS (RRMS), but not during progressive disease stages (Fig. 1g and Table 1). Together, these data demonstrate the expression of PD-L1 by astrocytes in response to acute autoimmune CNS inflammation and reveal increased levels of sPD-L1 in patients with CIS and RRMS.

### AhR and interferon signaling drive PD-L1 expression in astrocytes

To identify drivers of PD-L1 expression in astrocytes during autoimmune CNS inflammation, we stimulated primary mouse astrocytes with a range of mediators associated to an inflammatory milieu and measured Cd274 expression by RT-qPCR. TNF-α and IL-1β, two cytokines implicated in the pathogenesis of MS[30], as well as stimulation with the Type I interferon interferon β (IFN-β), and the Type-II interferon interferon γ (IFN-γ) induced Cd274 expression in astrocytes (Fig. 2a), which was also confirmed by flow cytometric analysis (Fig. 2b, c and Supplementary Fig. 2a, b). Interferon signaling has been described as a potent driver of PD-L1 expression in carcinomas[31], and peripheral application of IFN-β represents one of the first effective disease modifying therapies (DMT) approved for the treatment of MS. However, peripherally applied IFN-β does not readily cross the blood brain barrier (BBB)[32]. We have previously shown tissue-protective effects mediated by CNS-intrinsic Type I interferon signaling in astrocytes[12] and thus focused on the relevance of type-I interferons for the induction of astrocytic PD-L1. Stimulation with IFN-β strongly upregulated PD-L1 on mRNA and protein level on primary mouse and human astrocytes and further boosted PD-L1 expression under pro-inflammatory conditions

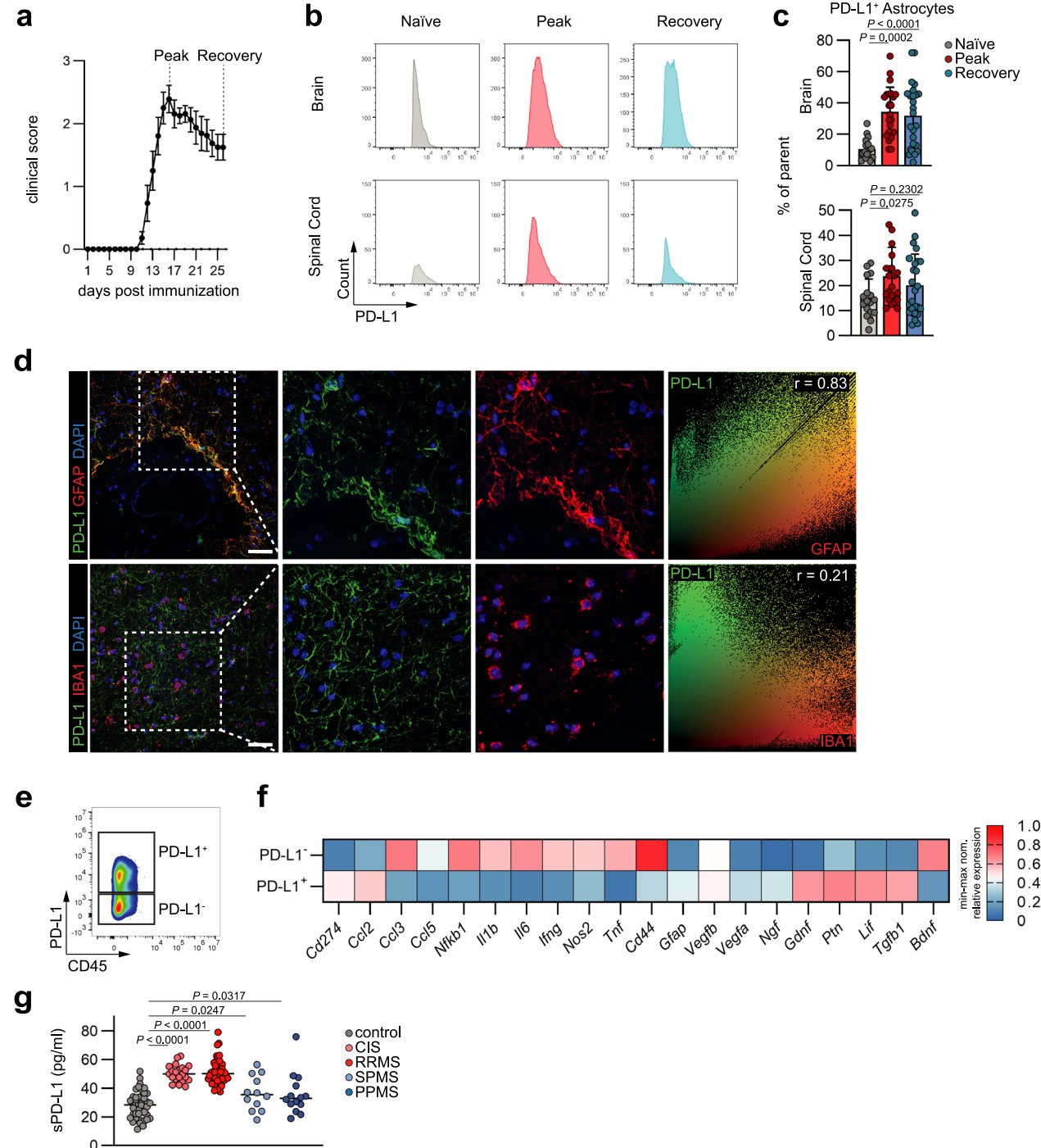

**Fig. 1 | Astrocytes express PD-L1 in the context of autoimmune CNS inflammation. a** Analysis of astrocytes over the course of EAE. Mice per group are $n = 5$ naive, $n = 6$ peak, $n = 8$ recovery. Experiment repeated three times. **b** Histograms depicting staining of PD-L1 on astrocytes in brain and spinal cord at the naïve, peak and recovery stage of EAE. **c** Relative expression (% of parent) of PD-L1 by brain and spinal cord astrocytes. $n = 17$ naive, $n = 24$ peak, $n = 25$ recovery. **d** Immunohistochemical staining of PD-L1+ astrocytes (GFAP+) and microglia (IBA1+) in active white matter lesions from MS patients. Data shown are representative of $n = 12$ fields from two distinct MS brains. 30 μm scale bar. Pearson's Correlation coefficient of global PD-L1/GFAP/IBA1 overlap. **e** Scatter plots of PD-L1+ astrocytes at peak of EAE sorted for mRNA analysis. **f** Normalized relative expression of genes in PD-L1+ and PD-L1- peak EAE astrocytes. $n = 3/10$ PD-L1+, $n = 3/10$ PD-L1-. **g** Quantification of soluble PD-L1 in cerebrospinal fluid from controls and patients with clinically isolated syndrome (CIS), relapsing-remitting MS (RRMS), secondary-progressive MS (SPMS), or primary-progressive MS (PPMS). $n = 44$ control, $n = 21$ CIS, $n = 53$ RRMS, $n = 12$ SPMS, $n = 14$ PPMS. One-way ANOVA with Dunnett's multiple comparisons test in not otherwise indicated. Multiple unpaired $t$ test in (**f**) with statistics provided in Supplementary Fig. 1d. Exact $P$ values are provided in the figure. Data shown as mean ± SD. Data shown as mean ± SEM in (**a**).

(Fig. 2d and Supplementary Fig. 2c−e), supporting our previous observations. This induction of astrocytic PD-L1 was reversed after treatment with Trichostatin A, a known suppressor of type-I interferon signaling (Fig. 2e). Additionally, Chromatin immunoprecipitation (ChIP) coupled to RT-qPCR revealed direct control of STAT1, the master transcription factor of IFN-signaling, following IFN-β stimulation at interferon-sensitive response elements (ISREs) in the PD-L1 promoter (Fig. 2f, Supplementary Fig. 2f, and Table 2).

**Table 1 | Patient characteristics of MS patients and controls**

| Cohorts (n) | Females (%) | Age (years) | Disease duration (years) | EDSS | Treatment |
|---|---|---|---|---|---|
| Fig. 1 | | | | | |
| Controls (44) | 31 (70.5%) | 37.1 [26.5: 48.3] | None | None | None |
| CIS (21) | 16 (76.2%) | 33.4 [27.1: 37.2] | None | 1.4 [0.5: 2.0] | None |
| RRMS relapse (53) | 36 (67.9%) | 33.2 [27.6: 38.5] | 2.3 [0.0: 3.0] | 1.9 [1.0: 3.0] | None |
| SPMS (12) | 19 (50.0%) | 55.3 [51.6: 63.1] | 14.2 [8.3: 17.0] | 5.2 [4.0: 6.9] | 3 (25.0%) |
| PPMS (14) | 6 (42.9%) | 47.2 [44.5: 54.8] | 4.8 [1.0: 6.5] | 4.0 [2.9: 4.5] | 1 (7.1%) |
| Fig. 2 | | | | | |
| CIS (17) | 13 (76.5%) | 34.1 [29.5: 37.0] | None | 1.4 [1.0: 2.0] | None |

Clinical information of patients used for the analysis of soluble PD-L1 in the cerebrospinal fluid and AhR activity by luciferase assay.

Finally, to in order to investigate the relevance of IFN-β as driver of astrocytic PD-L1 in vivo, we treated EAE mice with IFN-β by intranasal administration, a method we and others have previously shown to bypass the BBB[12,33]. Indeed, intranasal application of IFN-β upregulated *Cd274* together with other interferon induced genes in astrocytes, which was accompanied by an amelioration of EAE (Fig. 2g–i and Supplementary Fig. 2f–i). Together, these data demonstrate regulation of PD-L1 by type I and II interferons in astrocytes as well its therapeutic targetability by intranasal IFN-β administration.

Given the regulatory signature of PD-L1 expressing astrocytes in EAE, we aimed to define additional upstream regulators of PD-L1 in astrocytes. To that end, we used JASPAR[34] to predict transcription factor binding sites in the promoter region of the *Cd274* locus. Among Nf-κB and STAT1 binding sites, we found several binding sites of the aryl hydrocarbon receptor (AhR), a known regulator of protective astrocyte responses in the context of CNS inflammation[11,12,35] (Fig. 2j and Table 2). Indeed, stimulation with 3-indoxylsulfate (I3S), a tryptophan metabolite and AhR ligand, resulted in the upregulation of PD-L1 by mouse and human astrocytes, which could further be potentiated by the addition of TNF-α and IL-1β (Fig. 2k and Supplementary Fig. 2j). Furthermore, conditional deficiency of AhR in astrocytes during EAE (GFAP^Cre Ahr^fl/f), as described previously[11], was accompanied by a downregulation of *Cd274* on astrocytes (Supplementary Fig. 2k). ChIP analyses confirmed binding of AhR to AhR-binding sites in the *Cd274* promoter following stimulation with I3S (Fig. 2l and Supplementary Fig. 2l). In order to validate our observations and to compare AhR and interferon signaling as transcriptional drivers of *Cd274* expression quantitatively, we used a promoter reporter construct, in which activation of the *Cd274* promoter induces the expression of green fluorescent protein (GFP)[36]. Indeed, both IFN-β stimulation and a broad variety of AhR ligands activated the *Cd274* promoter in HEK293T cells (Fig. 2m). Collectively, this suggests that AhR signaling is a driver of PD-L1 expression during neuroinflammation.

To evaluate whether therapeutic induction of interferon or AhR signaling could facilitate the expression of *Cd274* by astrocytes in vivo, we induced EAE in WT mice and administered IFN-β, I3S, or vehicle intranasally daily starting at day 7 post immunization (Fig. 2n). Particularly IFN-β treatment significantly increased the expression of *Cd274* by both brain and spinal cord astrocytes and decreased disease severity, while I3S treatment predominantly increased *Cd274* expression by spinal cord astrocytes (Fig. 2o and Supplementary Fig. 2m). This was in line with a positive correlation between sPD-L1 levels in the CSF of MS patients and increased AhR activity (Fig. 2p and Table 1), indicating that microbial metabolites may contribute to immune checkpoint signaling in the CNS of MS patients. Finally, both interferon as well as AhR downstream genes were upregulated in PD-L1+ compared to PD-L1- astrocytes, supporting the notion that these pathways are associated to PD-L1 expression in astrocytes (Fig. 2q and Supplementary Fig. 2n). Overall, these data suggest that interferons and AhR ligands are strong drivers of astrocytic PD-L1 expression during CNS

inflammation in mice and humans. Since both IFN-β and AhR signaling in astrocytes have previously been linked to an anti-inflammatory and tissue-protective phenotype in the context of MS and EAE[11,12], this may indicate that PD-L1 signaling might contribute to these regulatory functions.

## Astrocyte-derived PD-L1 is important for the resolution of autoimmune CNS inflammation

To evaluate the overall relevance of PD-L1/PD-1 signaling in the context of autoimmune neuroinflammation, we induced EAE in WT mice and administered the small-molecule PD-L1/PD-1 checkpoint inhibitor BMS202 intranasally on a daily basis starting at day 7 post immunization (Fig. 3a). BMS202 has been described to dissociate PD-L1/PD-1 complexes by direct steric blockade of the PD-1 binding site, as well as facilitating the dimerization of PD-L1[37]. In vivo blockade of PD-L1/PD-1 signaling during EAE worsened disease severity and reduced recovery compared to vehicle treated mice (Fig. 3a, b). This was in line with an increase in pro-inflammatory monocytes, $T_H1$, and $T_H17$ cells in the CNS of BMS202-treated mice, while the number of anti-inflammatory IL-10 producing T cells was reduced (Fig. 3c and Supplementary Fig. 3a, b). Of note, we observed no differences in the number or proliferative capacity of T cells and monocytes in spleens of BMS202 treated mice compared to controls, suggesting that nasal BMS202 treatment primarily affects inflammatory processes in the CNS without major effects on splenic monocytes and T cells (Supplementary Fig. 3c–f).

Next, we sought to determine the contribution of astrocyte-derived PD-L1 to the anti-inflammatory effects of PD-L1 in the context of autoimmune CNS inflammation. For this, we abrogated *Cd274* expression in astrocytes using a lentiviral vector that co-expresses *Gfap*-driven CRISPR–Cas9 and a targeting single guide RNA (sgRNA)[6,38] (Fig. 3d and Supplementary Fig. 3g). Knockout of *Cd274* in astrocytes (*Gfap-Cd274*) worsened EAE and increased the expression of pro-inflammatory genes in astrocytes and microglia (Fig. 3d, e and Supplementary Fig. 3h, i). Moreover, high-dimensional flow cytometry, followed by dimensionality reduction and clustering further revealed an expansion of inflammatory cell clusters in *Gfap-Cd274* mice (Fig. 3f, g and Supplementary Fig. 4a, b). This was confirmed after manual gating and using SAM (significance analysis of microarray)[39], a statistical method developed to identify features from input data described by a response variable, demonstrating increased activation of microglia, as well as an increase in CD4+ T cells, B cells and neutrophils (Fig. 3h and Supplementary Fig. 4c–f).

To address the question whether astrocytic PD-L1 is contributing to the protective effects mediated by intranasal IFN-β as demonstrated in our previous experiments, we inactivated *Cd274* in astrocytes and treated the mice intranasally with IFN-β starting at day 7 post immunization. Indeed, inactivation of *Cd274* in astrocytes (Supplementary Fig. 4g) diminished the protective effects of IFN-β and worsened disease severity irrespective of the treatment, suggesting that the

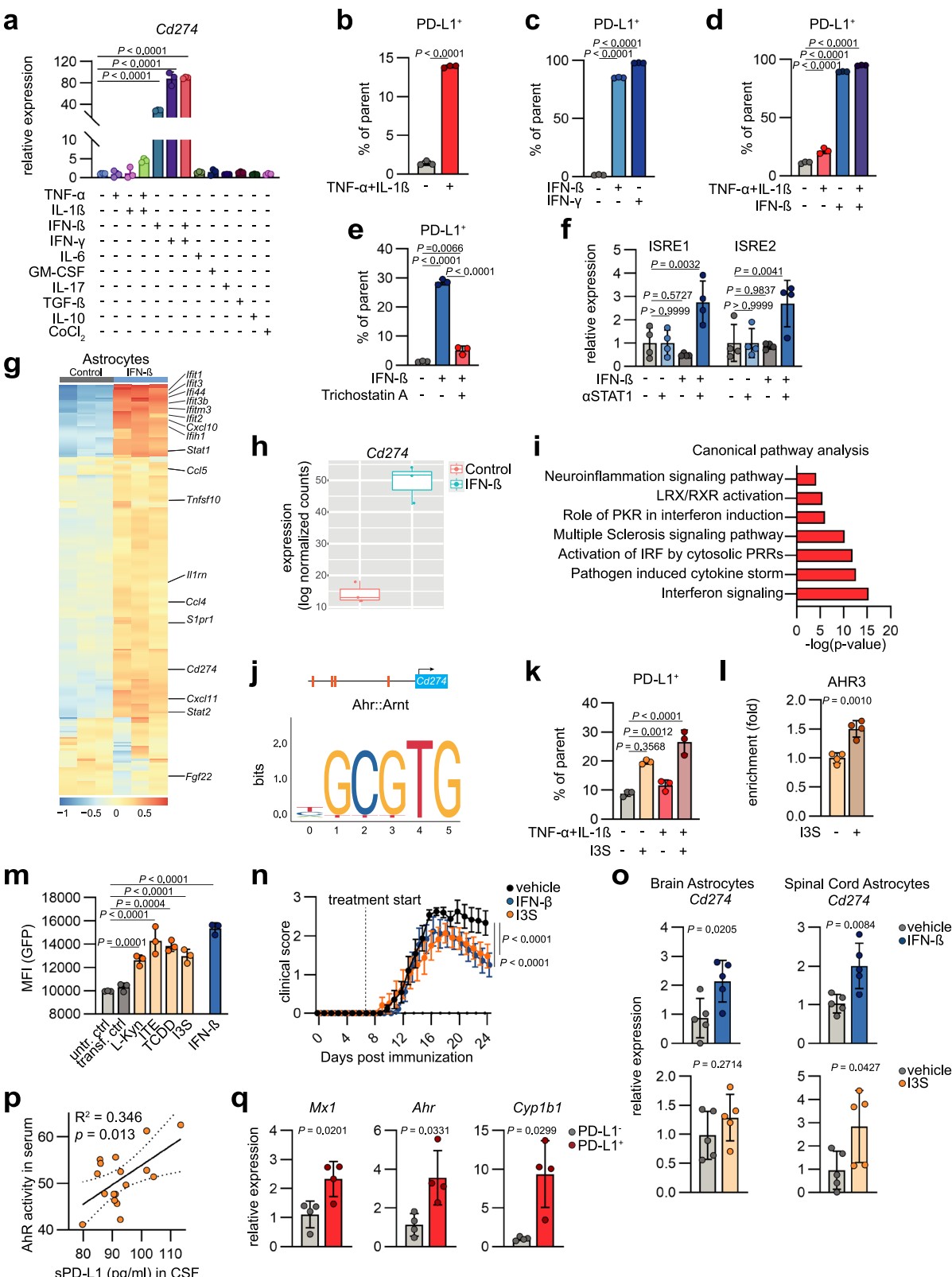

therapeutic potential of intranasal IFN-β administration is partially mediated by astrocyte-derived PD-L1 (Fig. 3i). Collectively, these data demonstrate that PD-L1 signaling in the context of autoimmune CNS inflammation is important for the control of autoimmune CNS inflammation and that astrocytic PD-L1 contributes to these functions by modulating the inflammatory potential of infiltrating and CNS-resident cell types.

## PD-L1/PD-1 signaling in chronic neuroinflammation regulates glial pathogenicity

To evaluate the relevance of PD-L1/PD-1 signaling in progressive stages of CNS inflammation, we induced EAE in WT animals and intranasally applied BMS202 daily as in our previous set of experiments, but now starting at peak of disease (day 16), once CNS intrinsic inflammation was established (Fig. 4a). Blockade of PD-L1/PD-1 signaling resulted in

Fig. 2 | **Astrocytic PD-L1 is induced by interferon and AhR signaling. a** Relative expression of *Cd274* in primary mouse astrocytes stimulated with pro- and anti-inflammatory stimuli. *n* = 3 per group. **b** Relative expression (% of parent) of PD-L1 in primary mouse astrocytes stimulated with TNF-α and IL-1β (*n* = 3) or vehicle (*n* = 3) measured by flow cytometry. Experiment repeated 3 times. **c**, relative expression (% of parent) of PD-L1 by primary mouse astrocytes following stimulation with IFN-β, IFN-γ, or vehicle measured by flow cytometry. *n* = 3 per group. **d** Relative expression (% of parent) of PD-L1 by human astrocytes following stimulation with TNF-α, IL-1β, and IFN-β measured by flow cytometry. *n* = 3 per group. **e**, relative expression (% of parent) of PD-L1 by primary mouse astrocytes following stimulation with IFN-β in combination with Trichostatin A or vehicle measured by flow cytometry. *n* = 3 per group. **f** ChIP-qCPR analysis of STAT1 recruitment to interferon-sensitive response elements (ISRE) in the *Cd274* promoter following stimulation with IFN-β or vehicle. *n* = 4 per group. **g** RNA-Seq analysis of sorted astrocytes from B6 mice treated with vehicle or IFN-β. *n* = 3 per group. **h** Expression (log normalized counts) of sorted astrocytes from C57Bl/6 mice treated with vehicle or IFN-β. *n* = 3 per group. **i** Ingenuity Pathway Analysis (IPA) of sorted astrocytes from vehicle and IFN-β treated mice. **j** Predicted AhR-binding sites in the *Cd274* promoter by JASPAR[80]. **k** Relative expression (% of parent) of PD-L1 by primary mouse astrocytes following stimulation with TNF-α, IL-1β, I3S, or vehicle measured by flow cytometry. *n* = 3 per group. **l** ChIP-qCPR analysis of AhR recruitment to the *Cd274* promoter following stimulation with I3S or vehicle, *n* = 4 per group. **m** Flow cytometric analysis of *CD274* promoter activation (depicted as median fluorescent intensity; MFI) following stimulation with inducers of AhR and interferon signaling using a promoter-GFP-reporter construct. *n* = 3 per group. **n** EAE development in mice intranasally treated with vehicle, IFN-β, or I3S starting at day 7 post immunization. *n* = 13 vehicle, *n* = 11 I3S, *n* = 9 IFN-β. Experiment repeated twice. **o** RT-qPCR analysis of *Cd274* expression in sorted astrocytes from brains and spinal cord tissue of mice treated with vehicle, IFN-β, or I3S throughout the course of EAE. *n* = 5 per group. **p** Linear regression analysis with 95 % confidence intervals of sPD-L1 (pg/ml) concentration in the cerebrospinal fluid (CSF) obtained from patients with clinically isolated syndrome (CIS) and their AhR activity measured by a promoter-reporter luciferase assay. *n* = 17. **q** RT-qPCR quantification of *Mx1*, *Ahr*, and *Cyp1b1* expression in PD-L1⁺ and PD-L1⁻ sorted astrocytes. *n* = 4 per group. Unpaired *t* test (two-tailed) in (**b**, **l**, **o**, **q**), One-way ANOVA with Dunnett's multiple comparisons test in (**c**–**f**, **k**, **m**), Area under the curve (AUC) of disease curves was used to determine statistical significance through *t* test with Dunnett's multiple comparisons test in (**n**). Exact *P* values are provided in the figure. Data are shown as mean ± SD if not indicated otherwise. Data are shown as mean with the 25th and 75th percentiles in (**h**). Data are shown as mean ± SEM in (**n**).

### Table 2 | Predicted transcription factor binding sites in the *Cd274* promoter

| Matrix ID | Name | Score | Relative score | Sequence ID | Start | End | Strand | Predicted sequence |
|---|---|---|---|---|---|---|---|---|
| MA0006.1 | MA0006.1.Ahr::Arnt | 8,069948 | 0,934962247 | | 21 | 26 | − | AGCGTG |
| MA0006.1 | MA0006.1.Ahr::Arnt | 8,069948 | 0,934962247 | | 136 | 141 | − | AGCGTG |
| MA0137.1 | MA0137.1.STAT1 | 16,045383 | 0,92387337 | | 428 | 441 | − | AAAAAACGAAACTA |
| MA0006.1 | MA0006.1.Ahr::Arnt | 7,6802692 | 0,918428643 | | 404 | 409 | + | GGCGTG |
| MA0137.1 | MA0137.1.STAT1 | 13,421521 | 0,869238698 | | 271 | 284 | − | TAAAAACGAAACTA |
| MA0006.1 | MA0006.1.Ahr::Arnt | 6,158047 | 0,853842612 | | 130 | 135 | − | TGCTTG |

Transcription factor binding sites have been predicted by JASPAR[80] using the 700 bp upstream region of the mouse and human *Cd274*/*CD274* locus with a 85% relative profile score threshold.

failure of recovery, concomitant with an increase in CD4⁺ T cells as previously observed following BMS202 treatment during acute stages (Fig. 4b). In line with this, we observed a reduction in regulatory T cells and increase in pro-inflammatory T cell subsets (Fig. 4c), suggesting PD-L1/PD-1 signaling controls T cell pathogenicity not only in acute but also progressive stages of neuroinflammation. This was complemented by an increase in pro-inflammatory cytokines produced by myeloid cells while their absolute cell numbers were not significantly changed following BMS202 treatment (Fig. 4d, e). In addition, BMS202 treatment during late stages of autoimmune CNS inflammation increased the production of pro-inflammatory cytokines by both microglia and astrocytes (Fig. 4f).

Next, we aimed to validate these observations in a more chronic model of autoimmune CNS inflammation. For this, we induced EAE in NOD/ShiLtJ mice and started intranasal treatment with BMS202 at their transgression into the chronic stage, which is characterized by progressive worsening of the disease. Similar to our observations in the context of acute CNS inflammation, PD-L1/PD-1 blockade exacerbated disease (Fig. 4g). Bulk RNA-seq of sorted microglia and astrocytes revealed an upregulation of pro-inflammatory genes by microglia (*Ccl5*, *Cxcl14*, *Cd74*, *C4*) and downregulation of tissue-protective genes by astrocytes (*Tgfb1*, *Ncan*, *Bcan*, *Gria1*) in BMS202-treated mice (Fig. 4h–k and Supplementary Fig. 5a–d). Interestingly, while both astrocytes and microglia showed an enrichment in pathways associated to lymphocyte activation and regulatory signaling (Fig. 4i and Supplementary Fig. 5b–d), effector T cells remained largely unaffected by BMS202 treatment (Supplementary Fig. 5e), suggesting that other cell types may be affected by these alterations. Finally, the increased inflammatory response of microglia following PD-L1/PD-1 blockade was confirmed by RT-qPCR, demonstrating the upregulation of pro-inflammatory cytokines in BMS202-treated mice (Fig. 4j, k).

Collectively, these data suggest that PD-L1/PD-1 signaling controls the inflammatory properties of peripheral immune cells but also microglia and astrocytes during chronic autoimmune CNS inflammation.

### Astrocytic PD-L1 controls inflammatory activities in PD-1⁺ microglia

Recent studies have described the expression of PD-1 in microglia and their regulation by PD-L1⁺ astrocytes in the context of Alzheimer's disease (AD) and other CNS insults[22,40,41]. To investigate whether microglia and other cell types can respond to PD-L1 in the context of autoimmune CNS inflammation, we induced EAE in WT mice and studied the spatiotemporal expression of PD-1 by high-dimensional flow cytometry (Fig. 5a, b). As expected, we observed an increase in PD-1⁺ T cells during onset and peak of disease, particularly in the spinal cord, recapitulating the disease dynamics in this acute model of autoimmune CNS inflammation (Fig. 5b). While pro-inflammatory T_H1 and T_H17 effector cells represented the dominant subsets among PD-1⁺ T cells at peak of disease, they reduced their expression of PD-1 in the recovery phase (Fig. 5c, d). In contrast, regulatory T cells retained high expression of PD-1⁺ throughout disease and represented the most abundant T cell subset during recovery stages, supporting previous reports that describe the control of effector regulatory T cells by PD-L1 and PD-1 signaling[42,43] (Fig. 5c, d).

In addition to T cells, microglia displayed high expression of PD-1 during onset, peak, and recovery stages both in the brain and spinal cord, matching the upregulation of PD-L1 by astrocytes (Fig. 5b and Supplementary Fig. 5f, g). Immunohistochemical analysis confirmed the presence of PD-1⁺ microglia in proximity to PD-L1⁺ astrocytes in the brain of EAE mice (Supplementary Fig. 5h), indicating that microglia may be responsive to astrocytic PD-L1. To evaluate whether the expression of PD-1 by microglia is functionally relevant in the context

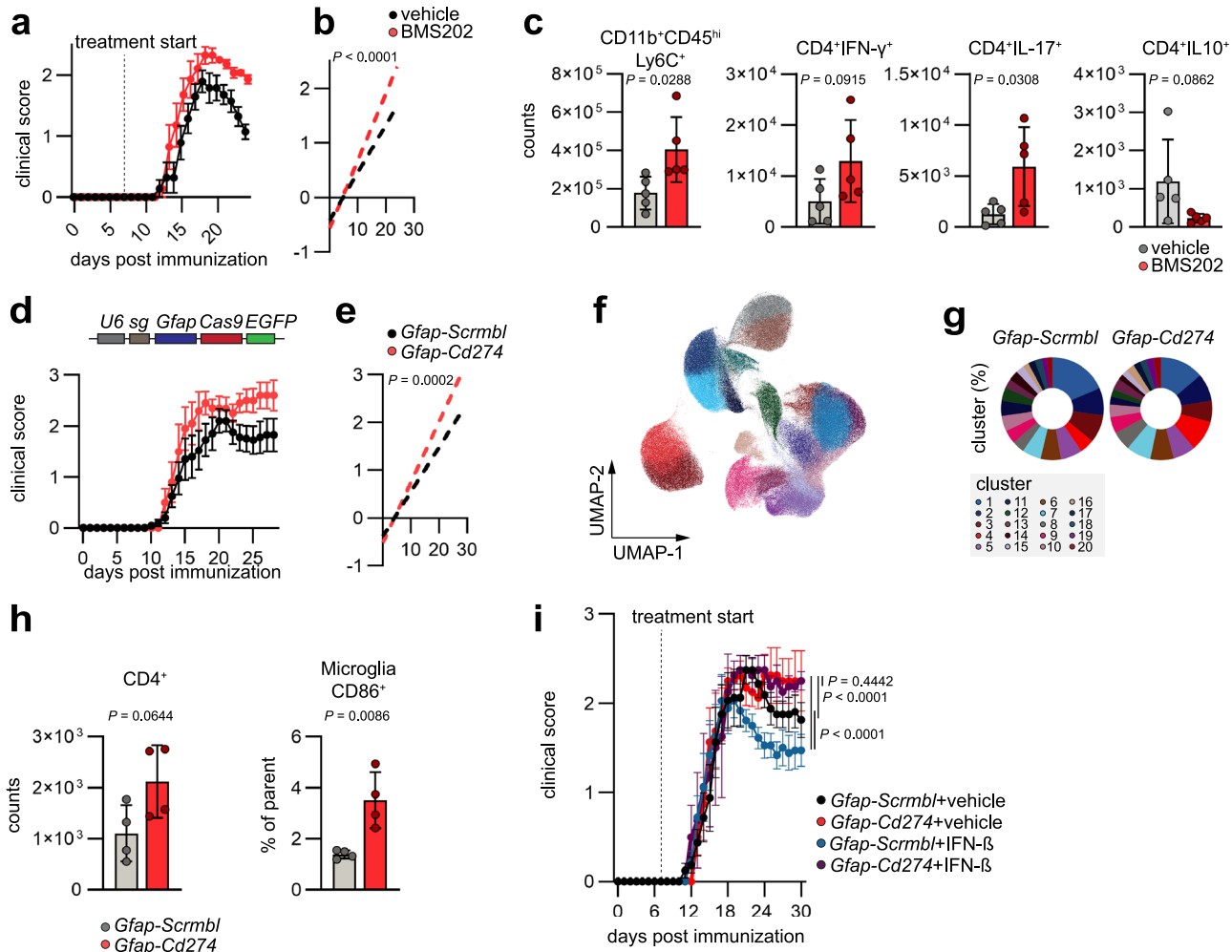

**Fig. 3 | PD-L1 expression by astrocytes limits autoimmune CNS inflammation.**
**a** EAE development and regression analysis (**b**) of clinical scores in mice intranasally treated with vehicle or BMS202 starting at day 7 post immunization. $n = 7$ vehicle per group. Experiment repeated twice. **c** Flow cytometric quantification of inflammatory monocytes (CD11b⁺CD45^hi Ly6C⁺), pro-inflammatory $T_H1$ (CD45⁺CD11b⁻CD4⁺IFN-γ⁺), $T_H17$ (CD45⁺CD11b⁻CD4⁺IL-17⁺), and regulatory IL-10⁺CD4⁺ T cells in the CNS (brain and spinal cord) of vehicle or BMS202 treated mice. $n = 5$ per group. **d** EAE development and regression analysis (**e**) in mice transduced with *Gfap-Scrmbl* ($n = 10$) and *Gfap-Cd274* ($n = 5$). Experiment repeated twice. **f** UMAP plots of CNS cells analyzed by high-dimensional flow cytometry in *Gfap-Scrmbl* ($n = 5$) and *Gfap-Cd274* ($n = 5$) mice. **g** Abundance of FlowSOM clusters

in the CNS of *Gfap-Scrmbl* ($n = 5$) and *Gfap-Cd274* ($n = 5$) mice analyzed by high-dimensional flow cytometry. **h** Quantification of microglia (CD45^int CD11b⁺) and CD4⁺ T cells (CD45⁺CD11b⁻) in the CNS of *Gfap-Scrmbl* and *Gfap-Cd274* mice analyzed by flow cytometry. $n = 4$ per group. **i** EAE progression in mice transduced with *Gfap-Scrmbl* and *Gfap-Cd274*, which were treated intranasally with vehicle or IFN-β starting at day 7 post immunization. $n = 8$ *Gfap-Scrmbl* + vehicle, $n = 9$ *Gfap-Scrmbl* + IFN-β, $n = 4$ *Gfap-Cd274* + vehicle, $n = 4$ *Gfap-Cd274* + IFN-β. Unpaired t test (two-tailed) in (**c**, **h**); Linear regression analysis in (**b**, **d**); Area under the curve (AUC) of disease curves was used to determine statistical significance through Tukey's multiple comparisons test in (**i**). Exact P-values are provided in the figure. Data are shown as mean ± SD in (**c**, **h**). Data are shown as mean ± SEM in (**a**, **d**, **i**).

of CNS inflammation, we inactivated *Pdcd1* under control of the *Itgam1* (*Cd11b*) promoter using a lentiviral CRISPR-Cas9 construct. Similar to *Cd274* inactivation in astrocytes, PD-1 abrogation in microglia exacerbated disease and reduced recovery (Fig. 5e and Supplementary Fig. 5i). This was accompanied by an increase in CD4⁺ T cells in the CNS and their expression of the pro-inflammatory cytokine IFN-γ (Fig. 5f, g). Furthermore, microglia and astrocytes upregulated the production of pro-inflammatory cytokines and tissue-degenerative inducible nitric oxide synthetase (iNOS) (Fig. 5h), collectively demonstrating that PD-1 signaling in microglia is required for the control of inflammatory processes in the context of neuroinflammation.

To test whether microglia are responsive to astrocytic PD-L1, and whether this interaction relies on physical or soluble cues, we first determined drivers of *Pdcd1* in primary mouse microglia. Similar to *Cd274* expression by astrocytes, microglia upregulated *Pdcd1* in response to TNF-α and IL-1β, as well as interferon signaling (Fig. 5i). Since IFN-γ was the strongest inducer of *Pdcd1* in microglia, we focused

on this stimulus for further investigation. Next, we tested the functional relevance of PD-L1/PD-1 -dependent astrocyte-microglia crosstalk in a primary co-culture setting. To that end, we separately activated astrocytes and microglia by stimulation with IFN-γ to induce the expression of PD-L1 or PD-1, respectively, followed by co-culture, and assessed their inflammatory phenotype by intracellular flow cytometry with or without the blockage of PD-L1/PD-1 signaling (Fig. 5j–k and Supplementary Fig. 5j). Since PD-L1 was also expressed by microglia (Fig. 1d, Supplementary Fig. 1b, and Supplementary Fig. 5g), we decided to block astrocyte-microglial PD-L1/PD-1 interactions by either inhibiting astrocytic PD-L1 through the addition of BMS202 prior co-culture or by blocking its binding to microglial PD-1 through the addition of an anti-PD-1 (αPD-1) antibody (Fig. 5j). Blockage of PD-L1/PD-1 interactions by both BMS202 and αPD-1 boosted the expression of pro-inflammatory markers (CCL-2, GM-CSF, iNOS) by microglia, while their effects on the proliferative and tissue-protective capacity of microglia were distinct, highlighting potential differences

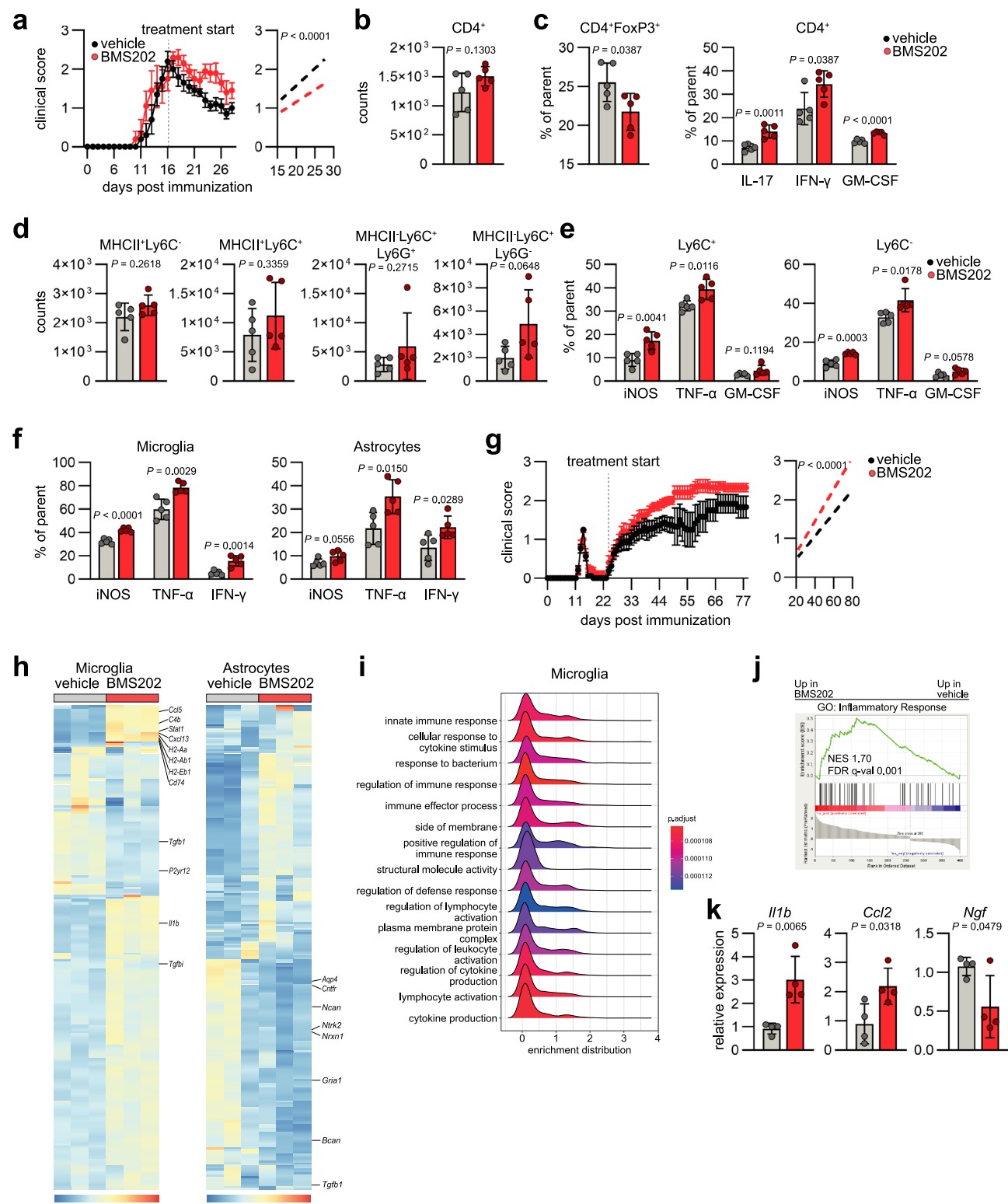

in their mechanism of action (MoA) (Fig. 5k). Of note, to exclude that PD-1 signaling via its second ligand programmed death ligand 2 (PD-L2) is the cause of the differences between blockade with BMS202 and αPD-1, we assessed the expression of PD-L2 in astrocytes following IFN-γ stimulation and observed no regulation, concomitant with previous reports that reported no relevance for PD-L2 in the context of neuroinflammation[44] (Supplementary Fig. 5k). Based on our observation that astrocytes upregulate the metalloproteases responsible for shedding PD-L1 during EAE (Supplementary Fig. 1f), we next sought to

determine whether soluble PD-L1 from astrocytes is sufficient to regulate the inflammatory potential of microglia. In order to do so, we induced PD-1 expression in microglia by IFN-γ stimulation and added astrocyte-conditioned medium (ACM) obtained from IFN-γ stimulated astrocytes, where PD-L1 was blocked by the addition of BMS202 or αPD-1 (Fig. 5l, m). Indeed, blockage of astrocytic PD-L1 by both BMS202 and αPD-1 polarized microglia towards a less protective, more pro-inflammatory phenotype, indicating that astrocyte-derived sPD-L1 may functionally regulate the pathogenic functions of activated

**Fig. 4 | PD-L1/PD-1 signaling in chronic autoimmune CNS inflammation limits glial pathogenicity. a** EAE progression and regression analysis in B6 mice treated intranasally with vehicle or BMS202 starting at peak of disease (day 16 post immunization). $n = 5$ vehicle, $n = 5$ BMS202. **b** Quantification of CD4$^+$ T cells (CD45$^+$CD11b$^-$) in the CNS of mice treated with vehicle or BMS202 during late stages of CNS inflammation. $n = 5$ per group. **c** Relative abundance (% of parent) of regulatory T cells (CD4$^+$FoxP3$^+$), T$_H$1 cells (CD45$^+$CD11b$^-$CD4$^+$IFN-γ$^+$), T$_H$17 cells (CD45$^+$CD11b$^-$CD4$^+$IL17$^+$) and GM-CSF producing cells (CD45$^+$CD11b$^-$CD4$^+$GM-CSF$^+$) in the CNS of mice treated with vehicle or BMS202 during late stages of CNS inflammation. $n = 5$ per group. **d** Quantification of CD45$^{hi}$CD11b$^+$MHCII$^+$Ly6C$^-$, CD45$^{hi}$CD11b$^+$MHCII$^+$Ly6C$^+$, CD45$^{hi}$CD11b$^+$MHCII$^-$Ly6C$^+$Ly6G$^+$, and CD45$^{hi}$CD11b$^+$MHCII$^-$Ly6C$^+$Ly6G$^-$ in the CNS of mice treated with vehicle or BMS202 during late stages of CNS inflammation. $n = 5$ per group. **e** Relative expression (% of parent) of pro-inflammatory cytokines iNOS, TNF-α, and GM-CSF produced by Ly6C$^+$ and Ly6C$^+$ myeloid cells in the CNS of mice treated with vehicle or BMS202

during late stages of CNS inflammation. $n = 5$ per group. **f** Relative expression (% of parent) of pro-inflammatory cytokines iNOS, TNF-α, and IFN-γ produced by microglia and astrocytes in the CNS of mice treated with vehicle or BMS202 during late stages of CNS inflammation. $n = 5$ per group. **g** EAE progression and regression analysis of NOD/ShiLtJ mice treated with vehicle or BMS202 starting at day 23 post immunization. $n = 16$ vehicle, $n = 16$ BMS202. Experiment repeated twice. **h** RNA-Seq analysis of sorted microglia and astrocytes from NOD/ShiLtJ mice treated with vehicle or BMS202 starting at day 23 post immunization. $n = 3$ per group. **i, j** pathway enrichment and GSEA (**h**) of microglial gene expression from NOD/ShiLtJ mice treated with vehicle or BMS202. $n = 3$ per group. **k** RT-qPCR analysis of *Il1b*, *Ccl2*, and *Ngf* expression by sorted microglia from NOD/ShiLtJ mice treated with vehicle or BMS202. $n = 4$ per group. Unpaired *t* test (two-tailed) in (**b–f, k**); Linear regression analysis starting day 16 p.i. in (**a**); Linear regression analysis starting day 23 p.i. in (**g**). Exact *P* values are provided in the figure. Data are shown as mean ± SD if not indicated otherwise. Data are shown as mean ± SEM in (**a, e**).

---

microglia (Fig. 5l, m and Supplementary Fig. 5l). Finally, to evaluate whether PD-L1 has therapeutic potential to modulate the inflammatory functions of microglia, we treated IFN-γ activated microglia with recombinant mouse PD-L1 (rmPD-L1) or vehicle and assessed their inflammatory phenotype by RT-qPCR (Fig. 5n). Inverse to our observations following blockade of soluble PD-L1 derived from astrocytes, the addition of rmPD-L1 suppressed the expression of proinflammatory genes and increased the expression of protective growth factors (Fig. 5n, o and Supplementary Fig. 5m), suggesting that treatment with recombinant PD-L1 in a neuroinflammatory setting may have the potential to modulate pathogenic glia responses. Altogether, these data identify microglial PD-1 as important checkpoint in the context of autoimmune CNS inflammation and demonstrate that soluble PD-L1 derived from astrocytes is a critical regulatory component that controls inflammatory signaling in microglia.

## Discussion

Immune checkpoint inhibitors represent one of the cornerstones of modern treatment against neoplastic diseases, where malignant cells take advantage of regulatory defense mechanisms to escape their extermination by immune cells. While the underlying mechanisms, including the PD-L1/PD-1 axis, have long been investigated in the context of T cell/APC crosstalk, it is becoming increasingly clear that also other cell types in the periphery, but also within the CNS have the capacity to regulate inflammatory responses through immunological checkpoints. In these lines, a series of studies in the context of tauopathy, aging, and AD suggests that co-regulatory signaling through PD-L1 and PD-1 may improve neuropathology and associated cognitive impairments, highlighting the relevance of this checkpoint in the CNS[23,45–47]. Moreover, new research demonstrates that glial cells can actively participate in this immune modulation via PD-L1/PD-1 signaling[20,48–51]. Here, we report that astrocyte-derived PD-L1 is critical for the resolution of autoimmune CNS inflammation in the context of MS and EAE and controls inflammatory properties of microglia.

First, we demonstrate that astrocytes are a major source of PD-L1 in the inflamed CNS, which is consistent with the literature that reports glial cell production of PD-L1 during neuroinflammation[22,50,52]. The expression of PD-L1 by astrocytes is strongly induced by IFN-γ, which is produced by infiltrating T$_H$1 and T$_H$17 cells in MS[53–55]. While IFN-γ exerts pathogenic activities during acute stages, recent reports have suggested that during chronic stages IFN-γ attenuates CNS inflammation and has the capacity to drive protective astrocyte subsets[13,56–59]. It is therefore reasonable to believe that IFN-γ induced PD-L1 expression by astrocytes contributes to the protective effects of IFN-γ, while its effect on other cell types may mediate its pathogenic functions. Similarly, IFN-β, a disease modifying drug used for the treatment of relapsing and secondary progressive MS, drives PD-L1 expression in astrocytes. We here show that the therapeutic effect of intranasal IFN-β is partially mediated through astrocytic PD-L1 expression, expanding our current

knowledge of how this treatment exerts its effects[60–62]. In addition to interferon signaling, we describe for the first time that AhR modulates astrocytic PD-L1 in the context of neuroinflammation. This is in line with a recent study that reports the control of immune checkpoint molecules in the tumor microenvironment by AhR[63]. Furthermore, AhR has previously been linked to protective roles in the context of MS where it limits pro-inflammatory signaling in astrocytes and microglia[11,12,64,65]. Together, these findings suggest that PD-L1 is part of a protective astrocyte signature that controls autoimmune CNS inflammation. Future studies will have to delineate whether these co-regulatory functions are exclusive to autoimmune neuroinflammation, or whether similar mechanisms exist in the context of low-grade innate inflammation seen in many primary neurodegenerative diseases.

In agreement with previous studies that used transgenic knockout or antibody-mediated inhibition of PD-L1 or PD-1 in the context of EAE[52,66,67], we demonstrate that intranasal administration of the small molecule PD-L1 inhibitor BMS202 worsens disease severity and increases the infiltration of pro-inflammatory monocytes and T cell subsets. While in the context of acute treatment paradigms this effect may be largely mediated through altered T cell responses, we show that blockade of PD-L1 during late-stage autoimmune CNS inflammation and a model of chronic MS drives the pathogenic activities of astrocytes and microglia. This is particularly relevant in regards to progressive MS, once ongoing inflammation and neurodegeneration occur behind a closed BBB and are predominantly driven by CNS-intrinsic processes.

Co-inhibitory signaling through PD-1 is a critical determinant of T cell functions and pathogenicity[18], and numerous studies describe protective effects of PD-1 expression by T cells during neuroinflammation[68–71]. In line with the existing literature, we demonstrate that PD-1 by effector T cells is induced in acute stages, and downregulated during late phases of EAE, while regulatory T cells maintain their high PD-1 expression throughout disease course. Indeed, the temporally distinct expression of PD-1 by different T cell subsets, similar to a recently described PD-1$^+$ T$_{eff}$/T$_{Reg}$ balance in the tumor microenvironment[72], may explain why early studies using antibody-mediated blockade of PD-1 during late stages of EAE observed no effect on disease development[52]. Furthermore, previous literature suggests that PD-L1 expressed by astrocytes and microglia has the capacity to suppress the activation of postencephalitic CD8 T$_{eff}$ cells and promotes their differentiation into long-lived brain resident CD8 memory cells in an in vitro co-culture setting[49,73], collectively strengthening the relevance of glial co-regulatory functions during neuroinflammation. While this has been shown in the context of virus induced CNS inflammation, the picture in MS is less clear. Here, a previous study suggests that while glial PD-L1 can functionally regulate CD8 T cell responses in vitro, the majority of infiltrating CD8 T cells in the lesion environment are insensitive to its inhibitory signal due to low expression of PD-1[50]. Instead, the authors observed high PD-1

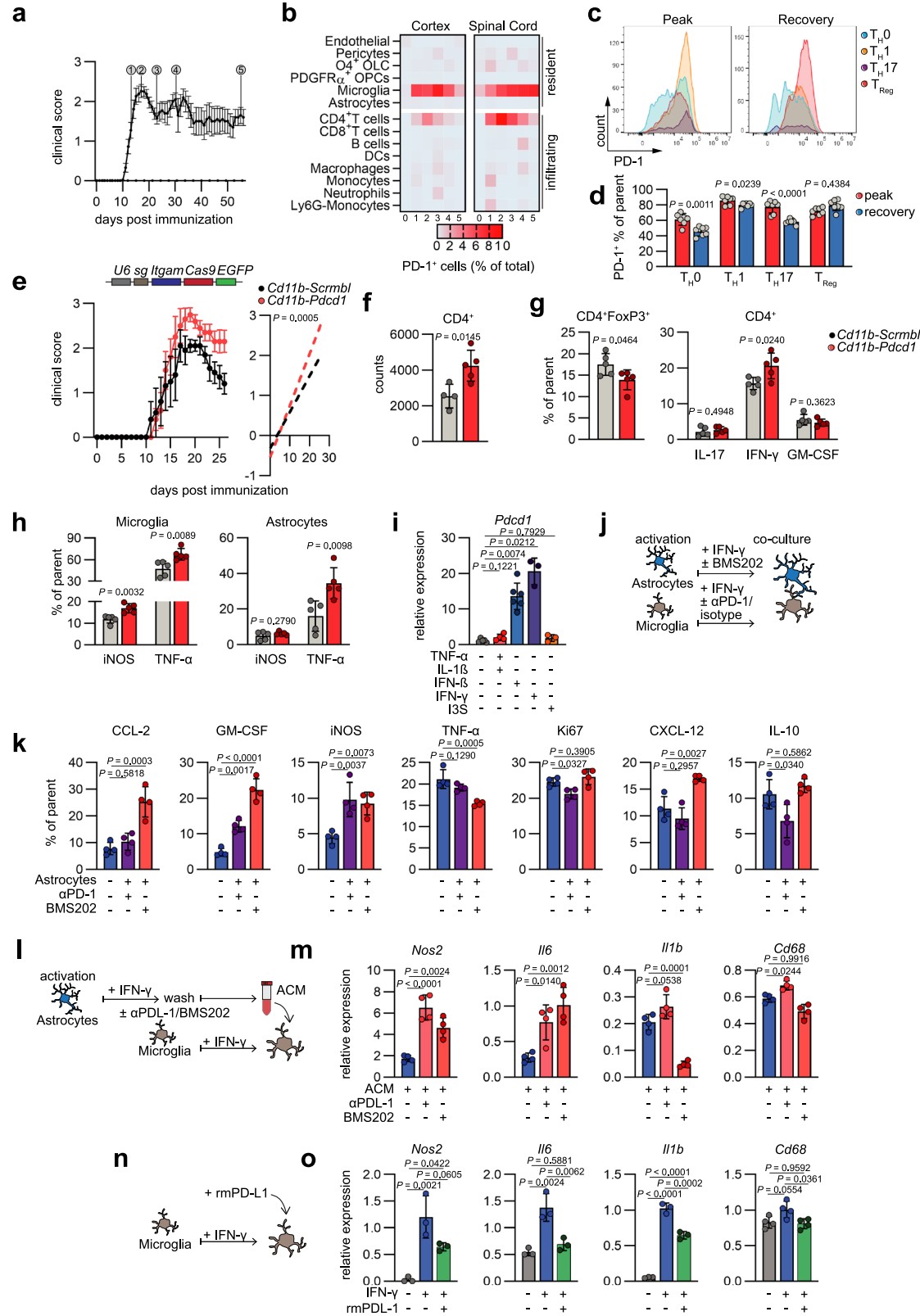

expression by CD4 T cells, matching our observations in EAE and supporting the relevance of glial PD-L1 in MS. Ultimately, future studies will have to delineate whether astrocytic PD-L1 is only a response to the inflammatory milieu induced by infiltrating immune cells or an effort to control and resolve the inflammation.

In addition to T cells, we detected high expression of PD-1 by microglia, an observation recently described in the context of AD and

brain injury[22,40,41]. In accordance with a high expression of microglial PD-1 particularly during late stages of CNS inflammation, we demonstrate that genetic abrogation of *Pdcd1* in microglia increases inflammatory signaling in microglia and astrocytes and limits recovery from EAE. In the light of a recent study demonstrating microglia PD-1 is important for their phagocytic capacity in the context of AD[22], it is conceivable that the increase in disease severity in *Cd11b-Pdcd1* mice is

**Fig. 5 | Astrocytic PD-L1⁺ attenuates pro-inflammatory signaling in microglia through PD-1. a** EAE development in B6 mice. 0 = naive, 1 = onset, 2 = peak, 3 = early recovery, 4 = late recovery, 5 = super late recovery. $n = 27$ **b** High-dimensional flow cytometric analysis of the cellular abundance in cortex and spinal cords throughout the course of EAE (**a**). $n = 3$ per timepoint and tissue. **c, d** Histograms of PD-1 staining and relative expression (% of parent) (**d**) in CD4⁺ T cell subsets in the CNS of EAE mice at peak and during recovery stages. $n = 7$ per group. **e** EAE development and regression analysis of mice transduced with *Cd11b-Scrmbl* ($n = 5$) and *Cd11b-Pdcd1* ($n = 5$). **f** Quantification of CD4⁺ T cells in the CNS of mice transduced with *Cd11b-Scrmbl* or *Cd11b-Pdcd1*. $n = 4$ *Cd11b-Scrmbl*, n = 5 *Cd11b-Pdcd1*. **g** Relative abundance (% of parent) of regulatory T cells (CD4⁺FoxP3⁺), $T_H1$ cells (CD45⁺CD11b⁻CD4⁺IFN-γ⁺), $T_H17$ cells (CD45⁺CD11b⁻CD4⁺IL17⁺), and GM-CSF producing cells (CD45⁺CD11b⁻CD4⁺GM-CSF⁺) in the CNS of mice transduced with *Cd11b-Scrmbl* or *Cd11b-Pdcd1*. $n = 5$ per group. **h** Relative expression (% of parent) of pro-inflammatory cytokines iNOS and TNF-α by microglia and astrocytes in the CNS of mice transduced with *Cd11b-Scrmbl* or *Cd11b-Pdcd1*. $n = 5$ per group. **i** RT-qPCR analysis of *Pdcd1* expression by primary mouse microglia following stimulation with vehicle ($n = 5$), TNF-α + IL-1β ($n = 4$), IFN-β ($n = 6$), IFN-γ ($n = 3$), or I3S ($n = 3$). **j** Schematic depicting the co-culture of primary mouse microglia and astrocytes with PD-1/PD-L1 blockade. **k** Relative expression (% of parent) of pro- and anti-inflammatory cytokines by primary mouse microglia following stimulation with IFN-γ and blockade of PD-L1 by BMS202, PD-1 by α-PD1 or isotype control measured by intracellular flow cytometry. $n = 4$ per group. **l** Schematic depicting the stimulation of primary mouse microglia with astrocyte-conditioned medium (ACM) from primary mouse astrocytes activated with IFN-γ and PD-L1 blockade by BMS202 or α-PD-L1. **m** RT-qPCR analysis of *Nos2*, *Il6*, *Il1b*, and *Cd68* by primary mouse microglia following blockade of soluble PD-L1 by BMS202 or α-PD-L1. $n = 4$ per group. **n** Schematic depicting the stimulation of primary mouse microglia with IFN-γ and recombinant mouse PD-L1 (rmPD-L1). **o** RT-qPCR analysis of *Nos2*, *Il6*, *Il1b*, and *Cd68* by IFN-γ-activated primary mouse microglia following treatment with rmPD-L1. $n = 3$ per group. Unpaired $t$ test (two-sided) if not indicated otherwise; Linear regression analysis in (**e**); One-way ANOVA with Tukey's multiple comparisons test in (**k, m, o**). Exact $P$ values are provided in the figure. Data are shown as mean ± SD in if not indicated otherwise. Data are shown as mean ± SEM in (**a, e**).

at least partially the result of an impaired clearance of myelin debris by microglia. We furthermore demonstrate that soluble PD-L1 produced by astrocytes is sufficient to suppress pro-inflammatory signaling in microglia, emphasizing the significance of astrocyte-microglial cross-talk in the context of CNS inflammation. The differences in effects observed after PD-L1 blockage by an anti-PD-L1 antibody or the small-molecule inhibitor BMS202 may be the consequence of their distinct MoA, as BMS202 has been shown to dissociate preformed PD-L1/PD-1 complexes and inhibits downstream signaling by inducing the formation of PD-L1 dimers, while antibody-mediated blockage likely sterically interferes with PD-L1 binding to PD-1.

In summary, we have identified PD-L1 expression by astrocytes as part of a protective activation state induced by interferon and AhR signaling. We show that astrocytic PD-L1 limits autoimmune CNS inflammation by suppressing pro-inflammatory functions of microglia. Overall, our findings define a co-regulatory mechanism between astrocytes and microglia that regulates acute and late-stage inflammation in the CNS.

## Methods

### Standard protocol approvals, registrations, and patient consent
This study was approved by the standing ethical committee (14/18S) at Technical University Munich. Written informed consent was obtained from every patient within the framework of the Biobank resources at the Department of Neurology at Technical University Munich, Germany, which is part of the national competence network Multiple Sclerosis. The animal studies were reviewed and approved by the standing ethical committees of the Bavarian State (Regierung von Oberbayern, AZ 55.2-2532.Vet_02-19-49; Regierung von Unterfranken, AZ 55.2.2-2532-2-1306).

### Mice
Mice were housed two to five animals per cage under a standard light cycle (12 h:12 h light:dark) (lights on from 07:00 to 19:00) at 20–23 °C and humidity (-50%) with ad libitum access to water and food. Adult female mice 8–12 weeks old and P0–P3 pups were used on a C57Bl/6J background (The Jackson Laboratory, #000664). NOD/ShiLtJ (The Jackson Laboratory, #001976) at 8–10 weeks of age were used for chronic EAE experiments.

### EAE
EAE was induced in 8-12-week female C57Bl/6J/NOD/ShiLtJ mice using 150 μg of MOG₃₅₋₅₅ (Genemed Synthesis, 110582) mixed with freshly prepared complete Freund's Adjuvant (using 20 ml Incomplete Freund's Adjuvant (BD Biosciences, #BD263910) mixed with 100 mg Myobacterium tuberculosis H37Ra (BD Biosciences, #231141)) at a ratio of 1:1 (v/v at a concentration of 5 mg/ml). All mice received two subcutaneous injections of 100 μl each of the MOG₃₅₋₅₅/CFA mix. All mice then received a single intraperitoneal injection of 200 ng pertussis toxin (List Biological Laboratories, #180) at a concentration in 200 μl of PBS. Mice received a second pertussis toxin injection at the same concentration two days after EAE induction. Mice were monitored and scored daily thereafter. EAE clinical scores were defined as follows: 0, no signs; 1, fully limp tail; 2, hindlimb weakness; 3, hindlimb paralysis; 4, forelimb paralysis; 5, moribund.

### Intranasal delivery of agents
Intranasal delivery of IFN-β, I3S, or vehicle (PBS) was started on day 7 after EAE induction. 20 μl of vehicle, 200.000 U/kg IFN-β, or 50 mg/kg I3S (all solved in PBS) was applied drop by drop on nostrils. Intranasal delivery of BMS202 in C57Bl/6J mice was initiated at day 7 post EAE induction or at peak of disease (day 16 post immunization). Intranasal delivery of BMS202 in NOD/ShiLtJ BMS202 was initiated at day 24 post EAE induction. In both cases, BMS202 was prepared at 10 mM in PBS and a total of 20 μl were applied on nostrils (3.81 mg/kg).

### Lentivirus production
Lentiviral vectors were produced as previously described[6,16,38,74]. Lentiviral vectors were obtained from lentiCRISPRv2 (Addgene #5296155), and lentiCas9-EGFP (Addgene #6359256). CRISPR/Cas9 lentiviral constructs were generated by modifying the pLenti-U6- *sgScramble-Gfap-Cas9-2A-EGFP-WPRE* lentiviral backbone. The *Gfap* promoter is the ABC₁D GFAP promoter. The *Itgam* promoter (also known as Cd11b) we described previously[11]. sgRNAs were substituted through a three-way cloning strategy using the following primers: U6-PCR-F 5′-AAAGG CGCGCCGAGGGCCTATTT-3′, U6-PCR-R 5′-TTTTTTGGTCTCCCGGTGT TTCGTCCTTTCCAC-3′, cr-RNA-F 5′-AAAAAAGGTCTCTACCG(sgRNA) GTTTTAGAGCTAGAAATAGCAAGTT-3′, cr-RNA-R 5′-GTTCCCTGCAGG AAAAAAGCACCGA-3. Products were amplified using Phusion Master Mix (Thermo Fisher Scientific Scientific, #F548S) and purified using the QIAquick PCR Purification Kit (Qiagen, 28104), followed by digestions using DpnI (NEB, #R0176S), BsaI-HF (NEB, #R3535/R3733), AscI (NEB, #R0558), or SbfI-HF (NEB, #R3642). Ligations were performed overnight at 16 °C using T4 DNA Ligase Kit (NEB, #M0202L). Ligations were transformed into NEB Stable Cells (NEB, #C3040) at 37 °C and single colonies were picked the following day. Plasmid DNA was isolated using QIAprep Spin Miniprep Kit (Qiagen, #27104) and the lentiviral plasmids were transfected into HEK293FT cells according to the Vira-Power Lentiviral Packaging Mix protocol (Thermo Fisher Scientific Scientific, #K497500) with pLP1, pLP2, and pseudotyped with pLP/VSVG. Medium was changed the next day, lentivirus was collected 48 h later and concentrated using Lenti-X Concentrator (Clontech, #631231) according to the manufacturer's protocol. Concentrates were resuspended in 1/100 of the original volume in PBS. Delivery of

lentiviruses via intracerebroventricular (i.c.v.) injection was performed as described previously[6,16,74]. In brief, mice were anaesthetized using 1% isoflurane mixed with oxygen. Heads were shaved and cleaned using 70% ethanol and Lidocain-gel followed by a medial incision of the skin to expose the skull. The ventricles were targeted bilaterally using the coordinates: ±1.0 (lateral), −0.44 (posterior), −2.2 (ventral) relative to Bregma. Mice were injected with approximately $10^7$ total IU of lentivirus delivered by two 10 µl injections using a 10 µl Hamilton syringe (Sigma-Aldrich, #20787) on a Stereotaxic Alignment System (Kopf, #1900), sutured, and permitted to recover in a separate clean cage. Mice received a subcutaneous injection of 1 mg/kg Meloxicam post i.c.v. injection and 48 h later. Mice were permitted to recover for between 4 and 7 days before induction of EAE. CRISPR–Cas9 sgRNA sequences were designed using a combination of the Broad Institute's sgRNA GPP Web Portal (https://portals.broadinstitute.org/gpp/public/analysis-tools/sgrna-design), Synthego (https://design.synthego.com/#/validate). sgRNAs used in this study were: Cd274: 5′- GTATGG-CAGCAACGTCACGA-3′; Pdcd1 5- GCTCAAACCATTACAGAAGG-3′; Scrmbl 5-GCACTACCAGAGCTAACTCA-3′.

### Primary mouse astrocyte and microglia cultures and stimulation experiments

Brains of mice aged P0–P3 were dissected into PBS on ice. Brains of 6-8 mice were pooled, centrifuged at $500 \times g$ for 10 min at 4 °C and resuspended in 0.25% Trypsin-EDTA (Thermo Fisher Scientific Scientific, #25200-072) at 37 °C for 10 min. DNase I (Thermo Fisher Scientific Scientific, #90083) was added at 1 mg/ml to the solution, and the brains were digested for 10 more minutes at 37 °C. Trypsin was neutralized by adding DMEM + GlutaMAX (Thermo Fisher Scientific Scientific, #61965026) supplemented with 10% FBS (Thermo Fisher Scientific Scientific, #10438026) and 1% penicillin/streptomycin (Thermo Fisher Scientific Scientific, #10500064), and cells were passed through a 70-µm cell strainer. Cells were centrifuged at $500 \times g$ for 10 min at 4 °C, resuspended in DMEM + GlutaMAX with 10% FBS 1% penicillin/streptomycin and cultured in T-75 flasks (Sarstedt, #83.3911.002), pre-coated with 2 µg/ml Poly-L Lysine (PLL, Provitro, #0413) at 37 °C in a humidified incubator with 5% $CO_2$ for 5–7 days until confluency was reached. Mixed glial cells were shaken for 30 min at 180 rpm, the supernatant was collected and the medium was changed, and then cells were shaken for at least 2 h at 220 rpm and the supernatant was collected and the medium was changed again. CD11b+ microglia were isolated from the collected supernatant using the CD11b Microbead Isolation kit (Miltenyi, #130-049-601) according to the manufacturer's instruction. For stimulation experiments, astrocytes and microglia were detached using TrypLE (Thermo Fisher Scientific, #12604013) and seeded in PLL-coated 48-well plates (Sarstedt, #NC1787625) at a density of 150.000 cells per well.

### Primary human astrocyte cultures and stimulation experiments

Primary human astrocytes were obtained from ScienCell (#1800) and cultured according to the manufacturer's instructions. In brief, cells were passaged in astrocyte medium (ScienCell, #1801) until confluency and subsequently plated onto plates pre-coated with 2 µg/ml Poly-L Lysine (Provitro, #0413). For stimulation experiments, astrocytes were detached using TrypLE (Thermo Fisher Scientific, #12604013) and seeded in PLL-coated 48-well plates (Sarstedt, #NC1787625) at a density of 150000 cells per well.

### Primary mouse co-culture and supernatant experiments

For co-culture experiments, astrocytes and microglia were activated with 20 ng/ml recombinant mouse IFN-γ (R&D, #485-MI-100/CF) overnight. The next day, microglia were detached using TrypLE and seeded onto primary mouse astrocytes with 1 µg/ml anti-PD-1 antibody (Biolegend, #135202) or isotype controls (Biolegend, #402301). After 24 h, cells were detached using TrypLE and intracellular flow

cytometry was performed. In case of supernatant experiments, astrocytes were activated with 20 ng/ml mouse IFN-γ in combination with or without 5 µM BMS202 for 24 h. After 24 h, medium was aspirated and fresh medium was added for another 24 h. After an additional 24 h, astrocyte-conditioned medium (ACM) was collected, cleared from debris by centrifugation and added onto activated microglia. After 24 h incubation, microglia were harvested for RNA isolation and RT-qPCR analysis.

### Cell culture experiments and stimulants

If not stated otherwise, the following concentrations were used for stimulation experiments: mouse recombinant mouse TNF-α (Peprotech, #AF-315-01A) 50 ng/ml, mouse IL-1β (Peprotech, #211-11B) 100 ng/ml, mouse IFN-β (R&D, #8234-MB-010/CF) 100 U/ml, IFN-γ (R&D, #485-MI-100/CF) 20 ng/ml, Trichostatin-A (Sigma, #T8552), I3S (Sigma, #I3875-250MG) 50 µg/ml, FICZ (Sigma, #SML1489-1MG) 1 µM, L-Kynurenin (Sigma, #K8625-25MG) 1 µM, TCDD (Sigma, #48599), ITE (Sigma, #SML3139), human TNF-α (Peprotech, #300-01 A) 50 ng/ml, human IL-1β (Peprotech, #200-01B) 100 ng/ml, human IFN-β (Peprotech, #300-02BC) 100 U/ml.

### Isolation of cells from adult mouse CNS

Mice were perfused with cold 1× PBS and the CNS was isolated and mechanically diced using sterile razors. Brains and spinal cords were processed separately or pooled (if not indicated otherwise) and transferred into 5 ml of enzyme digestion solution consisting of 35.5 µl papain suspension (Worthington, #LS003126) diluted in enzyme stock solution (ESS) and equilibrated to 37 °C. ESS consisted of 10 ml 10× EBSS (Sigma-Aldrich, #E7510), 2.4 ml 30% D(+)-glucose (Sigma-Aldrich, #G8769), 5.2 ml 1 M NaHCO3 (VWR, #AAJ62495-AP), 200 µl 500 mM EDTA (Thermo Fisher Scientific Scientific, #15575020), and 168.2 ml ddH2O, filter-sterilized through a 0.22-µm filter. Samples were shaken at 80 rpm for 30–40 min at 37 °C. Enzymatic digestion was stopped with 1 ml of 10× hi ovomucoid inhibitor solution and 20 µl 0.4% DNase (Worthington, #LS002007) diluted in 10 ml inhibitor stock solution (ISS). 10× hi ovomucoid inhibitor stock solution contained 300 mg BSA (Sigma-Aldrich, #A8806) and 300 mg ovomucoid trypsin inhibitor (Worthington, #LS003086) diluted in 10 ml 1× PBS and filter sterilized using a 0.22-µm filter. ISS contained 50 ml 10× EBSS (Sigma-Aldrich, #E7510), 6 ml 30% D(+)-glucose (Sigma-Aldrich, #G8769), and 13 ml 1 M NaHCO3 (VWR, #AAJ62495-AP) diluted in 170.4 ml ddH2O and filter-sterilized through a 0.22-µm filter. Tissue was mechanically dissociated using a 5-ml serological pipette and filtered through a 70-µm cell strainer (Fisher Scientific, #22363548) into a fresh 50-ml conical tube. Tissue was centrifuged at $600 \times g$ for 5 min and resuspended in 10 ml of 30% Percoll solution (9 ml Percoll (GE Healthcare Biosciences, #17-5445-01), 3 ml 10× PBS, 18 ml ddH2O). Percoll suspension was centrifuged at $600 \times g$ for 25 min with no breaks. Supernatant was discarded and the cell pellet was washed once with 1× PBS, centrifuged at $500 \times g$ for 5 min and prepared for downstream applications.

### Isolation of mouse splenic cells

Spleens were mechanically dissected and dissociated by passing through a 100-µM cell strainer (Fisher Scientific, 10282631). Red blood cells were lysed with ACK lysing buffer (Life Technology, A10492-01) for 5 min and washed with 0.5% BSA and 2 mM EDTA at pH 8.0 in 1× PBS and prepared for downstream applications.

### Flow cytometry

Live/dead staining was performed with LIVE/DEAD™ Fixable Aqua Dead Cell Stain Kit (Thermo Fisher Scientific Scientific, #L34957) according to the manufacturer's instructions. Cells were subsequently stained at 4 °C in the dark for 20 min with flow cytometry antibodies, diluted in FACS buffer (1× PBS, 2% FBS, 2 mM EDTA). Cells were then washed twice with FACS buffer and resuspended in 1× PBS for

acquisition. Antibodies used in this study were: BV421-CD11b (1:200; Biolegend, #101235), BV480-CD11c (1:100; BD, #565627), BV510-F4/80 (1:100; Biolegend, #123135), BV570-Ly6C (1:200; Biolegend, #128029), BV605-CD80 (1:100; BD, #563052), BV650-CD56 (1:100; BD, #748098), BV650-CD8 (1:100; Biolegend, #100741), PE-eFlour610-CD140a (1:100; Thermo Fisher Scientific, #61140180), SuperBright780-MHCII (1:200; Thermo Fisher Scientific, #78532080), BV711-CD74 (1:200; BD, #740748), PE-CD45R/B220 (1:100; BD, #561878), PE-CD105 (1:100; Thermo Fisher Scientific, #12-1051-82), PE-Ly6G (1:200; BioLegend, #127607), PE-CD140a (1:100; BioLegend, #135905), PE-O4 (1:100; Miltenyi, #130117507), PE-Ter119 (1:100; Biolegend, #116207), PE-Ly6C (1:100; Biolegend, #128007), AF488-A2B5 (1:100; Novus Biologicals, #FAB1416G), PE-CD279 (1:100; Biolegend, #135206), PE-CD273 (1:100; Biolegend, #107205), AF488-CD274 (1:100; Thermo Fisher Scientific, #53598282), PE-Cy5-CD24 (1:200; Biolegend, #101811), PE-Cy7-CD31 (1:200; Thermo Fisher Scientific, #25031182), PE-Cy7-CD274 (1:100; Thermo Fisher Scientific, #12598282), PerCP-eFlour710-CD86 (1:100; Thermo Fisher Scientific, #46086280), AF532-CD44 (1:100; Thermo Fisher Scientific, #58044182), PE-Cy5.5-CD45 (1:200; Thermo Fisher Scientific, #35045180), JF646-MBP (1:100; Novus Biologicals, #NBP2-22121JF646), APC-Cy7-Ly6G (1:200; Biolegend, #127623), AF700-O4 (1:200; R&D, #FAB1326N), BUV737-CD154 (1:100; BD, #741735), AF660-CD19 (1:100; Thermo Fisher Scientific, #606019380), APC/Fire810-CD4 (1:100; Biolegend, #100479). The following isotypes were used: AF488-Rat IgG2a Isotype Control (eBR2a) (1:100; Thermo Fisher Scientific, #53432180), PE-Cy7-Rat IgG2a Isotype Control (eBR2a) (1:100; Thermo Fisher Scientific, #25432182).

For intracellular flow cytometry staining, cells were fixed over night after surface staining using the Thermo Fisher Scientific™ Foxp3/Transcription Factor Staining Buffer Set (Thermo Fisher Scientific, #00552300) according to the manufacturer's instructions. For staining of intracellular cytokines, the following antibodies were used: PE-eFlour610-iNOS (1:100; Thermo Fisher Scientific, #61592080), BV711-IL17a (1:100; Biolegend, #506941), PE-CCL2 (1:100; BD, #554443), FITC-CXCL12 (1:100; Thermo Fisher Scientific, # MA523547), PE-Cy5-FoxP3 (1:100; Thermo Fisher Scientific, #15-5773-82), PE-Cy7-IFNγ (1:100; Biolegend, #505826), PE PerCP-eFlour710-TNF (1:100; Thermo Fisher Scientific, #46732180), APC-GM-CSF (1:100; Thermo Fisher Scientific, #17733182), AF700-Ki67 (1:100; BioLegend, #652419), APC-eF780-Ki67 (1:100; Thermo Fisher Scientific, #47569882), APC-eF780-IFNγ (1:100; Fisher Scientific, #47731942).

For fluorescence-activated cell sorting of astrocytes and microglia, CNS single-cell suspensions were prepared and stained as previously described[6,11,74]. In brief, all cells were gated on the following parameters: CD105negCD140anegO4negTer119negLy-6GnegCD45Rneg. Astrocytes were subsequently gated on: CD11bnegCD45negLy-6CnegCD11cneg. Microglia were subsequently gated on: CD11bhighCD45lowLy-6Clow. The antibodies used for fluorescence-activated cell sorting were: BV421-CD11b (Biolegend, #101235), BV570-Ly6C (Biolegend,# 128029), PE-CD140a (BioLegend, #135905), PE-O4 (Miltenyi, # 130117507), PE-Ter119 (Biolegend, #116207), PE-Ly6G (BioLegend, #127607), AF488-CD274 (Thermo Fisher Scientific, #53598282), PE-Cy7-CD274 (Thermo Fisher Scientific, # 12598282), PE-Cy5.5-CD45 (Thermo Fisher Scientific, #35045180). Compensation was performed on single-stained samples of cells and an unstained control. Cells were sorted on a FACS Aria IIu (BD Biosciences).

## Analysis of multiparameter flow cytometry data

Data was analyzed using the OMIQ platform. In brief, cells were gated as described previously[38,75] and subsampled to a set number of cells per group. Opt-SNE (max. 1000 iterations, perplexity 30, theta 0.5, verbosity 25) or UMAP (15 neighbors, minimum distance 0.4, 200 Epochs) was performed, followed by PhenoGraph clustering (based on Euclidean distance). SAM[39] was performed on groups using a two-class unpaired approach when two groups were compared (with max. 100 permutations and a FDR cutoff of 0.1).

## RNA isolation and RT-qPCR

Cells were lysed in 350 µl RLT buffer (Qiagen) and RNA was isolated using the RNeasy Mini Kit (Qiagen, #74004) according to the manufacturer's instructions. 500 ng RNA of each sample was transcribed into cDNA using the High-Capacity cDNA Reverse Transcription Kit (Life Technologies, #4368813). Gene expression was assessed by qPCR using the TaqMan Fast Advanced Master Mix (Life Technologies, #4444556). The following TaqMan-probes were used: Actb(Mm02619580_g1), Adam10(Mm00545742_m1), Adam17(Mm00456428_m1), Ahr(Mm00478932_m1), Aldh1l1(Mm03048957), Aqp4(Mm00802131_m1), Bdnf(Mm04230607_s1), Ccl2(Mm00441242_m1), Ccl3(Mm00441259_g1), Ccl5(Mm01302427_m1), Cd274(Mm03048248_m1), CD274(Hs00204257_m1), Cd44(Mm01277161_m1), Cd68(Mm03047343_m1), Cd86(Mm00444540_m1), Cyp1b1(Mm00487229_m1), Gapdh(Mm99999915_g1), Gdnf(Mm00599849_m1), Gfap(Mm01253033_m1), Gja1(Mm01179639_s1), Ifng(Mm01168134_m1), Il10(Mm01288386_m1), Il1b(Mm00434228_m1), Il6(Mm00446190_m1), Lif(Mm00434762_g1), Mmp13(Mm00439491_m1), Mmp9(Mm00442991_m1), Mx1(Mm00487796_m1), Nfkb1(Mm00476361_m1), Ngf(Mm00443039_m1), Nos2(Mm00440502_m1), Ptn(Mm01132688_m1), S100b(Mm00485897_m1), Tgfb1(Mm01178820_m1), Tnf(Mm00443258_m1), Vegfa(Mm00437306_m1), Vegfb(Mm00442102_m1).qPCR data were analyzed by the ΔΔCt method.

## Proliferation assay

Draining lymph node cells and splenocytes were isolated from MOG35-55 immunized mice that received intranasal BMS202 or vehicle at day 21 post immunization and processed as described before. In a 96 well flat bottom plate, 400,000 cells were re-stimulated for 72 with MOG35-55 (1 µg/ml, 10 µg/ml, 100 µg/ml). During the last 16 h, cells were pulsed with 1 µCi of $^3$[H]thymidine (PerkinElmer) followed by harvesting on glass fiber filters and analysis of incorporated $^3$[H] thymidine in a β-counter (1450 Microbeta, Trilux, PerkinElmer).

## Immunohistochemistry

For immunohistochemical analyses, mice were transcardially perfused with cold PBS. After perfusion with 4% PFA/1× PBS, the brain was dissected and processed for immunofluorescence labeling. The tissue was post-fixed in 4% PFA/1x PBS for 24 hours. After post-fixation, the brains were dehydrated at 4 °C in 30% sucrose in PBS overnight. By means of liquid nitrogen-cooled 2-methylbutane, the tissue was frozen in tissue-Tek embedding medium and kept at −20 °C for storage. 12 µm-thick cross cryostat sections (Leica) of brain tissue were obtained on glass slides and stored at −20 °C. Sections were then stained for GFAP, Iba1, PD-L1, and PD-1. Cross sections were incubated in acetone for 10 min at −20 °C for post-fixation. After washing the slides in 1× PBS for five minutes, they were incubated in blocking buffer (5% BSA/10% donkey serum/0.3% Triton-X/1x PBS) for 30 min. Slides were incubated overnight at 4 °C with mouse anti-GFAP (1:500; Millipore, #MAB360), rabbit anti-Iba1 (1:500; Abcam, #ab178846), rat anti-PD-L1 (1:200; Invitrogen, #14-5982-82) and goat anti-PD-1 (1:300; R&D, #AF1021) diluted in 1% BSA/1% donkey serum/0.3% Triton-X/1× PBS. On the following day, three washing steps of five minutes each preceded the incubation with the secondary antibodies for one hour: donkey anti-rat IgG AF405 (1:500; Abcam, ab175670), donkey anti-rabbit IgG AF568 (1:500; Life Technologies, #A10042), donkey anti-goat IgG AF647 (1:500, Life Technologies, #A32849) and donkey anti-mouse IgG AF488 (1:500; Life Technologies, #A21202). During the further procedure, sections were washed three times for five minutes. After this process, the slides were cover-slipped with Prolong Gold anti-fade and stored at 4 °C for further analysis.

Images of immunofluorescent labeled sections were acquired using the software Zen 3.0 (blue edition). Staining against GFAP, Iba1, PD-L1, PD-1 was examined using a fluorescence microscope (Axio Observer Z1, Zeiss) at 20x magnification. GFAP+ and Iba1+ cells were quantified manually in an unbiased manner by the same investigator. Image processing was performed using Photoshop CS6 (Adobe).

## Immunostaining of human brain tissue

Brain tissue was obtained from an untreated female individual with clinically diagnosed MS and neuropathologically confirmed MS, as previously described[12,76]. MS lesions were identified by demyelination and immune cell infiltration, as previously described[12,76]. All individuals, or their next of kin, had given informed consent for autopsy and use of their brain tissue for research purposes from the MS center (Centre de Resources et de Compétences Sclérose en Plaques Pays de La Loire) of the Nantes University Hospital (ABM PFS13-003 "Collection sclérose en plaques"). MS samples were processed and immunostained as previously described[5]. Briefly, sections were thawed, fixed, washed and blocked with 10% donkey serum. Sections were then incubated overnight at 4 °C with antibodies against PD-L1 (1:300; Biolegend, #329701). After washing, the samples were incubated at room temperature for 40 min with the secondary antibody Alexa-Fluor-488-conjugated donkey anti–mouse IgG (1:500; Jackson Immunoresearch labs, #715545150). Sections were then incubated with anti-GFAP antibody that was directly conjugated with Cy3 (1:300; Sigma Aldrich, #C9205) for 1 h. Imaging was performed using a Leica SP5 confocal microscope and the Leica LAS AF software. Images were processed using Adobe Photoshop CS2. For imaging analysis all of the data were acquired using the same settings, which were originally standardized on normal appearing white matter (NAWM) sections. The degree of colocalization of PD-L1 with GFAP was determined using the Leica LAS software. The overlap coefficient is expressed as a percentage, where 100% represents the maximum degree of colocalization, and 0% denotes no colocalization.

## Enzyme-linked immunosorbent assay

For the analysis of soluble PD-L1 in cerebrospinal fluid (CSF) of MS patients and controls, a commercial PD-L1 ELISA Kit was used (Invitrogen, #BMS2212) according to the manufacturer's instruction. CSF was obtained from the Joint Biobank Munich in the framework of the German Biobank node. For Patient characteristics see Supplementary Table 1. "Age", "Disease duration" and "EDSS" are shown as mean, with 25% and 75% percentiles indicated in square brackets. "Treatment" indicates the absolute number and percentage of treated patients. No additional relevant comorbidities or pharmaceutical treatments were reported in patients or controls.

## AhR and CD274 promoter activity assay

HEK293 cells were used in the transient transfection system, as previously described[64]. In brief, 20,000 cells per well were plated in 96-well flat-bottom plates. For the assessment of AhR agonistic activity, cells were transfected with pGud-Luc21 and pTK-Renilla (Renilla luciferase under control of constitutively active thymidine kinase promoter, Promega, Madison, WI) using Fugene-HD Transfection Reagent (Promega, #E2311) as suggested by the manufacturer 24 h after plating. After 24 h, transfected cells were incubated with DMEM supplemented with 10% of patient serum in duplicates. Luciferase activity was analyzed 24 hours later using the Dual Luciferase Reporter System (Promega, #E1980). Firefly luciferase activity was normalized to Luciferase activity to determine relative AhR activity. Results are represented as percent normalized AhR activity relative to control samples.

For the assessment of *CD274* promoter activity, cells were transfected with pEZX-PF02 (eGFP under control of *CD274* promoter) using Fugene-HD Transfection Reagent (Promega, #E2311) as suggested by the manufacturer. After 24 h, transfected cells were stimulated with the indicated stimuli. After 24 h stimulation, cells were harvested and the promoter activation indicated by the GFP signal was analyzed by conventional flow cytometry.

## Chromatin immunoprecipitation

Astrocytes were treated for 1 h followed by cell preparation adopted from the ChIP-protocol provided by Abcam. In brief, one confluent 150 mm² dish of cells was fixed in 0.75% formaldehyde for 10 min with gentle rotation, followed by adding glycine to a final concentration of 125 mM for 5 min. Subsequently, cells were washed two times with cold 1× PBS, and then scraped in 1× PBS. Cells were pelleted and resuspended in ChIP lysis buffer for 10 min on ice. Samples were sonicated, cell debris were pelleted and supernatant was used for immunoprecipitation. Each sample was diluted 1:10 with RIPA buffer (Thermo Fisher Scientific, #89900). Primary antibodies were added to the sample for 1 h at 4 °C. Sheared chromatin was immunoprecipitated with prepared A/G beads according to the Abcam protocol overnight at 4 °C with rotation. The next day, the protein-bound magnetic beads were washed 1× with low salt buffer, 1× with high salt buffer, and 1× with LiCl buffer. 120 μl Elution buffer was added to the samples for 15 min while vortexing gently. 5 M NaCl and RNase A was added to the sample according to the protocol and incubated while shaking overnight at 65 °C. Proteinase K was added for 1 h at 60 °C. DNA was purified using QIAquick PCR Purification Kit (Qiagen, #28104). qPCR was performed using Fast SYBR Green Master Mix (Thermo Fisher Scientific, #4385612). Anti-IgG immunoprecipitation and input were used as controls. We used rabbit anti-AHR (1:2000; Enzo Life Sciences, #BMLSA210), and rabbit IgG polyclonal isotype control (1:2000; Cell Signaling, #2975 S). PCR primers were designed with Primer3[77] to generate 50–150-bp amplicons. Primer sequences used were:

AHR1-F:5´-GGGGCTCTGAACTCGAGATA-3´, AHR1-R: 5´-CGCGA-GAAGATGCTCCCTTA-3´, AHR2-F:5´-CGGTTGTCTTGGAGCTCTCT-3´, AHR2-R: 5´-TTCCTGCGGATGACTTTAGAG-3´, AHR3-F:5´-GATCCCTG-GACCACGCTG-3´, AHR3-R: 5´-ATTTTGGGTGGGAGTGGAAC-3´, AHR 4-F: 5´-ATCCACAGCGTTCACAAAGG-3´, AHR4-R: 5´-CGTGGATTTGGT GACTTCCTC-3´.

## Bulk RNA-seq

Library preparation for bulk-sequencing of poly(A)-RNA was done as described previously[78]. Briefly, barcoded cDNA of each sample was generated with a Maxima RT polymerase (Thermo Fisher) using oligo-dT primer containing barcodes, unique molecular identifiers (UMIs) and an adapter. 5′-Ends of the cDNAs were extended by a template switch oligo (TSO) and full-length cDNA was amplified with primers binding to the TSO-site and the adapter. NEB UltraII FS kit was used to fragment cDNA. After end repair and A-tailing a TruSeq adapter was ligated and 3′-end-fragments were finally amplified using primers with Illumina P5 and P7 overhangs. In comparison to Parekh et al.[78], the P5 and P7 sites were exchanged to allow sequencing of the cDNA in read1 and barcodes and UMIs in read2 to achieve a better cluster recognition. The library was sequenced on a NextSeq 500 (Illumina) with 59 cycles for the cDNA in read1 and 18 cycles for the barcodes and UMIs in read2. Data was processed using the published Drop-seq pipeline (v1.0) to generate sample- and gene-wise UMI tables[79]. Reference genome (GRCm38) was used for alignment. Transcript and gene definitions were used according to the GENCODE version M25. Differential expression analysis was performed using R and DESeq2 (1.38.0). A *P* value of <0.05 was used to determine differentially expressed genes. The data

## Prediction of transcription factor binding sites

Putative transcription factor binding sites were identified using JASPAR[80] (https://jaspar.genereg.net/) on the 700 bp upstream region of the mouse and human *Cd274/CD274* locus with a 85% relative profile score threshold.

## Spatial transcriptomics

Visium spatial transcriptome plots were obtained from https://liddelowlab.shinyapps.io/GliaSeqPro/ with original data from Hasel et al.[9].

## Statistics and reproducibility

GSEA or GSEAPreranked analyses were used to generate enrichment plots for bulk RNA-seq using MSigDB molecular signatures for canonical pathways. For pre-ranked GSEA, GSEA software by the Broad Institute[81,82] was used and genes were ranked based on logFC and adjusted p-value. Ridgeplots were built using the Python v.3.5.0 package clusterProfiler[83] v.4.4.4. All samples were randomly allocated into treatment groups. To identify differentially expressed pathways, IPA software (Qiagen) was used by inputting gene expression datasets with corresponding log(FoldChange) expression levels with adjusted $P$ value < 0.05 compared to other groups. Canonical pathways metrics were considered significant at $P < 0.05$. All other statistical examinations were carried out using GraphPad Prism 9 (v. 9.5.1).

For analysis of multiple groups, 1-way ANOVA with Tukey's or Dunnett's multiple comparisons test was applied. For multiple testing, a 2-way ANOVA with Sidak's multiple comparisons test was applied. Family-wise significance and confidence level was set at $P < 0.05$. Additional information on the study design, the number of replicates, and the statistical tests used are provided in the figure legends.

## Reporting summary

Further information on research design is available in the Nature Portfolio Reporting Summary linked to this article.

## Data availability

Sequencing data have been deposited into the Gene Expression Omnibus (GEO) under the SuperSeries accession number GSE239875. Spatial Transcriptomic data obtained from Hasel et al. is deposited under the Gene Expression Omnibus SuperSeries accession number GSE148612. Clinical data for patient samples can be found in Supplementary Table 1. All other data that support the findings of this study are available from the corresponding author on request. Source data are provided with this paper.

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

## Acknowledgements

This work was supported by grants RO4866-3/1, RO4866-4/1 from the Deutsche Forschungsgemeinschaft (DFG). M.L. and V.R. were funded by an ERC Starting Grant by the European research Council (HICI 851693). V.R. was supported by a Heisenberg fellowship and Sachmittel support provided by the German Research Foundation (Deutsche Forschungsgemeinschaft, DFG, RO4866-3/1, RO4866-4/1) as well as in transregional and collaborative research centers provided by the German Research Foundation (DFG, Project ID 408885537 - TRR 274; Project ID 261193037-CRC 1181). T.B. was funded by the German Research Foundation (Deutsche Forschungsgemeinschaft, DFG, RO4866-3/1, RO4866-4/1). T.T. was funded by the Kommission für Klinische Forschung (KKF), Klinikum rechts der Isar. L.L. was funded by transregional research center provided by the German Research Foundation (DFG, German Research Foundation - Project ID 408885537 - TRR 274). F.J.Q. received funding from the National Institutes of Health, by the NMSS and the Progressive MS Alliance (NS087867, ES025530, ES032323, AI126880 and AI149699). T.K. was supported by the Deutsche Forschungsgemeinschaft (SFB1054-B06 (ID 210592381), TRR128-A07 (ID 213904703), TRR128-A12 (ID 213904703), TRR128-Z02 (ID 213904703), TRR274-A01 (ID 408885537), and EXC 2145 (SyNergy, ID 390857198), the ERC (CoG 647215), and by the Hertie Network of Clinical Neuroscience. B.H. received funding for the study by the European Union's Horizon 2020 Research and Innovation Program (grant MultipleMS, EU RIA 733161), the Deutsche Forschungsgemeinschaft (DFG, German Research Foundation) under Germany's Excellence Strategy within the framework of the Munich Cluster for Systems Neurology (EXC 2145 SyNergy—ID 390857198) and the Clinspect-M consortium funded by the BMBF. B.H. is associated with DIFUTURE (Data Integration for Future Medicine, BMBF 01ZZ1804[A-I]). The Biobank of the Department of Neurology as part of the Joint Biobank Munich in the framework of the German Biobank Node supported the study. We thank all members of the Rothhammer laboratory for helpful advice and discussions. We thank Ulrike J. Naumann for technical assistance and all the patients and their families that agreed to participate in this study.

## Author contributions

M.L. and T.B. performed most in vitro and in vivo experiments. L.L., D.F., O.V., A.P., T.T., and L.N. assisted with in vitro and in vivo experiments. L.L. performed immunohistochemical analyses of mouse tissue. H.K., D.L., and J.I.A. performed immunohistochemical analyses of human tissue. R.O. performed RNA sequencing, T.E. and R.R. gave input to bioinformatical analyses. M.L., D.F., and V.R. wrote the manuscript with input from coauthors. M.L., T.B., and V.R. designed the study and edited the manuscript. J.I.A., T.K., B.H., and F.J.Q. provided reagents and human samples. V.R. supervised the study.

## Funding

## Competing interests

The authors declare no competing interests.
