## [Peer review file · Nature Communications]

REVIEWER COMMENTS

Reviewer #1 (Remarks to the Author):

In the present study, the authors suggest that PD-L1 expression by astrocytes in animal models of EAE and in MS can potentially be protective.

While the overall topic is interesting, some of the conclusions are overstated, and some of the explanations are confusing.

General comments:

Throughout the manuscript, the authors refer to "inflammation" without specifying the type of inflammation to which they refer, and this use is imprecise, given that inflammation associated with EAE is different from that observed in other brain conditions, such as Alzheimer's disease and even other forms of MS. In addition, in the discussion, the authors should discuss their results in the context of the recently reported effect of PD-1/PD-L1 blockade in aging and in Alzheimer's disease.

Several specific issues should also be addressed:

1. The authors should be more specific when discussing cell type(s) expressing PD-L1.
2. It is well documented that PD-L1 is upregulated by different cell populations in the context of tissue inflammation or neuroinflammation, and that it undergoes activity-mediated shedding from the cell membrane. Specifically, in the context of glial cells (microglia, astrocytes) PD-L1 expression in neuroinflammation (Lokensgard et al., 2015, *Glia*).

3. In the present study, the authors isolated PD-L1+ and PD-L1- astrocytes from brain tissue and suggest that: "PD-L1+ astrocytes were associated with a protective signature and showed a reduction in pro-inflammatory gene expression, indicating that PD-L1+ astrocytes represent a tissue-protective reactive astrocyte subtype during autoimmune CNS inflammation." This is based on combined analysis of sorted astrocytes (brain and spinal cord pooled) "at the naïve (n = 4), peak (n = 4) and recovery (n = 4) stage of EAE". Figure 1e shows combined data of all astrocytes collected. As the authors themselves note, the inflammatory state of the issue is completely different in each one of these stages of EAE. While in Figure 1b-d, the analysis takes this into account, it is not clear why in Figure 1e all the astrocytes are pooled together.

Given the data it is not clear whether the PD-L1 expression is induced by the neuroinflammation rather than the PD-L1 expression is part of the attempt to resolve the inflammation.

4. The authors state that the "protective signature" of the PD-L1+ astrocytes is based on the genes *Ccl5*, *Nos2*, *Cd274*, *Gfap*, *Tgfb1*, *Lif*, *Ptn*, *Gdnf*, and *Ngf*. It is not clear why these genes and not others? The authors should clearly explain how these genes were chosen. There are many genes that are more classically associated with a pro- or anti-inflammatory response; why were these not selected? This is important since the conclusion that "PD-L1+ astrocytes are neuroprotective" drives the entire narrative, and the experimental basis for this conclusion is not explained.

5. Related to this, the part of the study showing the effect of astrocytic conditioned medium on microglia is interesting, but it was not unequivocally demonstrated that it is the sPD-L1 secreted into the medium that affects the microglia.

6. The difference in Figure 5o between IFN and IFN+BMS202 is not significant for TNF and IL-6 upregulation. Also, the genes tested here are not the same genes as those in the "protective signature" described at the beginning of the study. Why were different genes chosen for this part of the work?

7. BMS202 is not introduced in the Methods. The only information given is that it is "small-molecule PD-L1 / PD-1 checkpoint inhibitor". What is the formulation/preparation that was used for the

intranasal delivery? The authors should add a detailed description in the Methods.

8. Figure 3c. This referee could not find any description in the manuscript text, or the legends stating from what tissue the immune cells were isolated. It appears that these cells came from brain tissue, but this must be explicitly noted.

Reviewer #2 (Remarks to the Author):

In this manuscript, the authors report on the regulation of chronic inflammatory processes by astrocytes via the immune checkpoint PD-L1/PD-1. This mechanism is investigated through the elegant combination of genetic perturbation studies and pharmacological approaches, both in vivo and in vitro. The authors show that PD-L1 expression is driven by aryl hydrocarbon receptor (AhR) and interferon signaling, and they ultimately propose the glial PD-L1/PD-1 axis as novel therapeutic target for the modulation of CNS intrinsic mechanisms relevant to progressive MS. The study is novel and relevant, yet there are several points the authors need to address:

- Fig. 1: The conclusion that PDL1+ astrocytes are reparative based on the fact that they show lower expression of Ccl5 and Nos is an over statement. In fact, the gene profile shown in Fig 2 is not sufficient to support the conclusion that this population has a suppressed inflammatory phenotype. These conclusions should be toned down.
- Fig. 1: it is essential that the authors show PDL1 expression also in the normal brain to be able to claim that increased astrocytic PDL1 correlates with disease.
- Fig. 1: the authors show that sPDL1 is up in the CSF. Because in mice the authors measure the membrane bound form of PDL1, which is elevated, and sPDL1 is not measured, to better support the claim of correlation between human and mouse data, it would be important to show that expression/activity of metalloproteinases responsible of PDL1 cleavage is upregulated at peak EAE.
- Fig. 2: there are inconsistencies in the way the data are presented: why is PDL1 expression after stimulation with different cytokines represented as % population in one case and MFI in another? Both % population and MFI should be reported in all cases, as they address different points, cell frequency and expression levels.
- Fig. 2: the fact that multiple STAT1-binding sites were identified in the Cd274 promoter by JASPAR is not confirmation that Trichostatin A, a suppressor of type-I interferon signaling, abolished the induction of PDL1 expression by astrocytes. It opens that possibility, which still remains to be directly proven. This needs to be addressed or toned down.
- PDL1 expression is increased by TNF/IL1b but to a much lower extent than with IFNs, both I and II. Furthermore, it's specifically IFNg, not TNF or IL1b that seems to be the inflammatory stimulus pertinent to the MS environment that increases PDL1. There is selectivity in PDL1 expression based on these data. The authors should elaborate better on this point.
- Fig. 3A: the EAE experiment in Fig. 3A is not convincing. The EAE severity is minimal, especially in the vehicle control where the scores barely go above 1 (no locomotor phenotype) and essentially go down to 0 within 10 days. This is odd and seems to indicate a failure of EAE. In fact, EAE in the treated group seems in line with what is expected in an untreated scenario. Ultimately, it doesn't look like the treatment made it worse, but that the EAE induction failed or was suboptimal. The experiment should be repeated.
- The fact that T cells in the periphery do not change is not an indication that "... nasal BMS202 treatment primarily affects inflammatory processes in the CNS without major peripheral effects". Immune cells are trafficking more into the CNS, as shown by the authors (Fig. 3C), which may be due to increased T cell proliferation and activation in the periphery prior to entering the CNS. Also, other myeloid populations should be assessed as well as B cells.
- It is unclear what the significance of the Ly6C+ and Ly6C- myeloid population is, what proportion of the peripheral myeloid population these are, and why they are specifically singled out.
- In several occasions, results are overinterpreted and conclusions overstated. For example, in results

line 220: "Bulk RNA-seq of sorted microglia and astrocytes revealed increased pathogenic activity....": expression changes of a few genes from a bulk sequencing experiment cannot be taken as indication of "increased pathogenic activity". These statements need to be toned down.

Minor:

Please check for typos and incorrect wording throughout. For example, legend Fig. 1d: the MFI diagrams are not histograms, please correct.

Reviewer #3 (Remarks to the Author):

Key results

This article aims at investigating the immune regulation and tissue-protective functions of reactive astrocytes in the context of autoimmune CNS inflammation, by focusing of the PD-L1/PD-1 signaling. Authors show that PD-L1 is upregulated in astrocytes in the acute phase of EAE in mice and in acute inflammatory lesions on MS patient brain sections. Authors showed that PD-L1 is a downstream effector of both IFN β and AhR signaling. Using cell type specific approaches to knockout PD-L1 gene in astrocytes and PD-1 gene in microglia, authors showed a worsening of EAE phenotype and increased inflammatory mechanisms. Pharmacological blockade of PD-L1/PD-1 signaling worsens disease progression, suggesting that this signaling could be protective in both acute and progressive phases of EAE. A last sets of in vitro experiments suggest that PD-L1 can be secreted from astrocytes to bind PD1 and decrease the expression of pro-inflammatory cytokines in microglia.

Validity

Most in vivo manipulations of the PD-L1/PD-1 signaling do not use cell-type specific approaches (only experiments with lentiviral vectors) and because the astrocyte-specific upregulation of PD-L1 is not convincingly showed (see first § in the Significance section), it is possible that in EAE, activated microglia (and possibly other cell types) also upregulate PD-L1 and that PD-1 acts as an endocrine ligand. More importantly, there is no validation whatsoever that the CRISPR-Cas9-mediated knockout of PD-L1 gene in astrocytes and PD-1 gene in microglia after icv injections of lentiviral vectors approach is efficient and specific.

The timing of PD-L1 expression in astrocytes and of PD-1 in microglia over the course of EAE does not appear consistent. The number of PD-1+ microglia is maximal at recovery after EAE (Fig 5b) whereas PD-L1+ astrocytes levels are comparable to naïve mice at that time point (Fig 1c). Thus, it is not clear how this result can be reconciled with the author's conclusion that PD-L1 signaling in astrocytes also influence the progressive phases in EAE.

Some shortcuts are made between the interpretation of in vitro and in vivo experiments (line 56-57). For example, line 92: "concomitant" with should be replace by "these results are consistent with...."as it is just a parallel.

There are instances were result interpretations were not supported by significant data.

-Line 148, "IFN β and I3S significantly increased the expression of Cd274 by brain and spinal cord astrocytes": not true in spinal cord astrocytes, especially, because one outlier increases the mean Cd274 levels in the IFN β condition

-Line 254 "expression of pro-inflammatory cytokines": only IFN γ is significantly different, not IL-17 and GM-CSF

-Figure 5m The PD1 blocking antibody does not lead to significant decrease in PD1 expression

-Line 269 "suggesting that astrocytic PD-L1 controls pathogenic activities in microglia" this conclusion would have been true if authors used PD-L1 antibody instead of PD-1 antibodies

Significance

The main novelty is the attempt at dissecting astrocyte PD-L1 and microglia PD-41 crosstalk, but the *in vivo* experiments either using cell type isolation via flow cytometry and through genetic loss of function approaches are not fully convincing. In my opinion, authors included a significant amount of experiments using non cell type specific approaches (IFN β treatment etc) both *in vivo* and *in vitro*, which replicate the team's previous findings, to the expense of additional experiments to convincingly demonstrate that astrocytes play a key role in the protective effects of PD-L1/PD-1 signaling in EAE.

Experiments on progressive stages of EAE are interesting.

Detection of PD-L1 in astrocytes in acute inflammatory lesions on brain sections from MS patients has been described previously although the reference is not mentioned by authors (Pittet et al. The majority of infiltrating CD8 T lymphocytes in multiple sclerosis lesions is insensitive to enhanced PD-L1 levels on CNS cells. *Glia*. 2011 May;59(5):841-56)

Data and methodology

Some controls experiments are missing

Examples:

- Validation of cell type specific isolation using flow cytometry (Figure 1b-c). Is there a negative selection for neurons?
- Figure 1c: What about PD-L1 expression level in isolated microglia? This is an important control as there are reports that PD-L1 levels can also be expressed in microglia
- Immunofluorescent stainings quality is poor: negative control (no primary antibody) for PD-L1? It is a bit surprising that a membrane-bound protein perfectly co-localizes with filamentous GFAP.

There are few instances of where tool validation is limited or absent

- No validation of successful targeting of astrocytes and microglia using icv injection of lentiviral vectors is missing, despite the construct carrying eGFP, which could allow identifying transduced cells.
- Line 141: There is no mention of the reference paper using these reporters (ref 56). Authors should mention that these experiments were performed *in vitro* and not in primary cell cultures of astrocytes but in HEK293T cells.

There is a lack of clarity when explaining the rationale for some experiments. In addition, the precise experimental approach is often not clearly mentioned in the results text (method, type of samples etc).

Examples:

- Line 67: how PDL1+ astrocytes were identified? Authors should mention in the main text that it was done by flow cytometry (as opposed to immunostainings)
- Legend of Supp Fig 1a n=10 PD-L1 (samples)? From how many mice?
- Which CNS region is used (brain or spinal cord), would be useful to add more details (which brain region? Which spinal cord level?) or justify that EAE-mediated inflammation is widespread and does not show prominent regional heterogeneity (grey vs white matter for example)
- How MS lesion was identified? Co-staining with immune cell markers? Demyelination?
- Line 114-115: "mouse and human astrocytes" authors should mention from primary cell cultures
- Line 118: the rationale is not clearly stated. Authors should be explicit "because IFN primarily signals through STAT1..."
- Fig2f: relevance? Statistical analysis? There are probably many other transcription factor-binding

sites. This result might not be critical to show, rather move to supplemental and replace by plot in SuppFig2g showing cd174 upregulation.

-Supp Fig 4 h: why use LPS? These results are not presented or interpreted.

Figure and plot size

Overall, the figures and plots are very small. Authors should consider removing p value numbers and replace by stars to show significance.

Analytical approach

Your assessment of the strength of the analytical approach, including the validity and comprehensiveness of any statistical tests. If any aspect of the analytical approach is outside the scope of your expertise, please note this in your report or in the comments to the editor.

Sampling size and composition is often not clear. For example, for cell culture experiments, it is stated that n=3/condition and that the experiments were repeated 3 times. Were the data averaged?

Some stats are missing Figure 2 "o" p value for the comparison IFN β (written INF β) and I3S groups.

Suggested improvements

- A schematic/graphical abstract summarizing the main findings would be useful

- Figure 1c: Additional RT-qPCR for astrocyte markers would be required to validate the isolation by flow cytometry

-Line 76: Explain how these specific genes were chosen? (Fig 1f, easier to read if in the order as in SuppFig1a). Did authors check the expression anti-inflammatory genes?

-It would be interesting to determine whether PD-L1 is differentially upregulated in different types of lesions (as done in Schirmer et al. 2014)

-On all clinical score plots: only half of the error bars (upwards on the top curve and downwards on the bottom curve) which is misleading. Both error bars should be shown in all plots.

- Gender considerations: Experiments in EAE mice were performed in females only but not justified (add reference to the literature). For histological analysis of human brain samples from controls and MS patients, gender is not provided

Minor comments:

- Line 172 and 174: Knockout rather than inactivation or knockdown

- Line 268 add "in microglia"

Figure 2 d in legend replace "mouse" by "human" astrocytes

Figure 2 h not useful without gene names

Figure 2 m legend missing

Figure 2 k vehicle tag in black, not grey

Figure 2 o and n should be swapped to improve readability

- §line 195 Rather belongs in the introduction, cut is short in the results

- Line 370 GFA-ABC1D or gfa2 which one?

- § line 505: missing to mention PDL1 antibody for FACS isolation of astrocytes?

- § line 519: primers or sequences for other genes used throughout the manuscript?

- Line 570: Normal appearing white matter abbreviated only

Clarity and context

Some words or concept are not introduced, described or explained

Examples:

- Line 121 "ISGF3" not explained
- Line 135: GFAP-cre; AChfl/fl mouse model description is missing from the material and method section. In addition, cd174 levels are not significantly decreased. The downregulation is mild but visually enhanced on the plot because the axis is cut
- Line 141: There is no mention of the reference paper using these reporters (ref 56). Authors should mention that these experiments were performed in vitro and not in primary cell cultures of astrocytes but in HEK293T cells.
- §line 190: The result interpretation concerning the recovery phase in the EAE mouse model is not straightforward

References

Pittet et al. The majority of infiltrating CD8 T lymphocytes in multiple sclerosis lesions is insensitive to enhanced PD-L1 levels on CNS cells. *Glia*. 2011 May;59(5):841-56 should be mentioned

Your expertise

I am not an immunologist so parts related to peripheral immune responses

REVIEWER COMMENTS

Reviewer #1 (Remarks to the Author):

In the present study, the authors suggest that PD-L1 expression by astrocytes in animal models of EAE and in MS can potentially be protective.

While the overall topic is interesting, some of the conclusions are overstated, and some of the explanations are confusing.

General comments:

1. Throughout the manuscript, the authors refer to "inflammation" without specifying the type of inflammation to which they refer, and this use is imprecise, given that inflammation associated with EAE is different from that observed in other brain conditions, such as Alzheimer's disease and even other forms of MS.

We thank the Reviewer for this important point. In order to clearly delineate the type of inflammation referred to, we have adjusted the wording throughout the manuscript and now explicitly refer to "autoimmune" inflammation whenever appropriate.

2. In addition, in the discussion, the authors should discuss their results in the context of the recently reported effect of PD-1/PD-L1 blockade in aging and in Alzheimer's disease.

We thank the reviewer for this comment. Indeed, recent research has shown the relevance of PD-1 / PD-L1 on both the inflammatory and degenerative component e.g. in the context of AD, and aging¹⁻⁴. In order to put our novel findings on autoimmune inflammation in relation and context to these observations, we have added a section in the introduction and discussion in the revised version of our manuscript highlighting the relevance of PD-L1 / PD-1 checkpoint signaling in both primarily degenerative and autoinflammatory CNS disorders.

3. The authors should be more specific when discussing cell type(s) expressing PD-L1. It is well documented that PD-L1 is upregulated by different cell populations in the context of tissue inflammation or neuroinflammation, and that it undergoes activity-mediated shedding from the cell membrane. Specifically, in the context of glial cells (microglia, astrocytes) PD-L1 expression in neuroinflammation (Lokensgard et al., 2015, Glia).

We appreciate this helpful comment by the Reviewer. In the revised version of the manuscript, we have now added quantification of other major PD-L1 expressing cell types during autoimmune inflammation in the EAE model (Extended Data Fig. 1a-b). Furthermore, we have also included the reference by Lokensgard et al. brought forward by the reviewer in the main text.

4. In the present study, the authors isolated PD-L1+ and PD-L1- astrocytes from brain tissue and suggest that: "PD-L1+ astrocytes were associated with a protective signature and showed a reduction in pro-inflammatory gene expression, indicating that PD-L1+ astrocytes represent a tissue-protective reactive astrocyte subtype during autoimmune CNS inflammation." This is based on combined analysis of sorted astrocytes (brain and spinal cord pooled) "at the naïve (n = 4), peak (n = 4) and recovery (n = 4) stage of EAE". Figure 1e shows combined data of all astrocytes collected. As the authors themselves note, the inflammatory state of the issue is completely different in each one of these stages of EAE. While in Figure 1b-d, the analysis takes this into account, it is not clear why in Figure 1e all the astrocytes are pooled together.

We appreciate the opportunity to clarify this point. For the analysis of PD-L1⁺ vs. PD-L1⁻ astrocytes we sorted cells from pooled brain and spinal cord tissue at peak of EAE, as we observed the highest upregulation of PD-L1 on astrocytes during this stage irrespective of their origin. To make this more clear, we have adapted the figure legend and provided additional information in the main text.

6. Given the data it is not clear whether the PD-L1 expression is induced by the neuroinflammation rather than the PD-L1 expression is part of the attempt to resolve the inflammation.

This is an important point raised by the Reviewer and a rather philosophical to answer. In our opinion, our data shows that PD-L1 expression by astrocytes is present in inflamed CNS tissue, but only very low in naïve mice. Hence, we argue that the expression of PD-L1 by astrocytes is associated to an inflammatory environment and thus induced by neuroinflammation, a reasoning that is also supported by the upregulation of PD-L1 by primary mouse astrocytes following stimulation with various cytokines, for which we have included additional data in the revised version of this paper (Fig. 2a). Moreover, the lack of astrocytic PD-L1 exacerbates EAE, and blockage of PD-L1 in supernatant derived from astrocytes boosts pro-inflammatory processes in microglia, overall suggesting that PD-L1 exerts regulatory effects in this context and thus contributes to the resolution of inflammation. Together, while one might not be able to directly answer the question of hen and egg here, we now discuss this point in the discussion section of the revised manuscript.

7. The authors state that the “protective signature” of the PD-L1⁺ astrocytes is based on the genes Ccl5, Nos2, Cd274, Gfap, Tgfb1, Lif, Ptn, Gdnf, and Ngf. It is not clear Why these genes and not others? The authors should clearly explain how these genes were chosen. There are many genes that are more classically associated with a pro- or anti-inflammatory response; why were these not selected? This is important since the conclusion that “PD-L1⁺ astrocytes are neuroprotective” drives the entire narrative, and the experimental basis for this conclusion is not explained.

We thank the Reviewer for highlighting this important point and agree that the limited selection of genes does not allow such statement. We therefore toned down the respective description. Furthermore, we have repeated the experiment and now show a total of 20 genes relevant to an inflammatory state of reactive astrocytes (Fig. 1f, Extended Data Fig. 1d). The majority of genes includes soluble factors that either mediate protection or drive autoimmune inflammation^{5,6}. In addition, we now also show activation markers relevant to astrocyte biology (Fig. 1f, Extended Data Fig. 1d). While there is no clear definition of protective versus inflammatory astrocyte phenotypes, we are confident that based on this extension suggested by the Reviewer, we now are able to show that PD-L1 positive astrocytes belong to a more anti-inflammatory, tissue-protective subtype of astrocytes.

8. Related to this, the part of the study showing the effect of astrocytic conditioned medium on microglia is interesting, but it was not unequivocally demonstrated that it is the sPD-L1 secreted into the medium that affects the microglia.

This is an important point raised by the Reviewer. We agree that this claim cannot be made by solely using the anti-PD-L1 inhibitor BMS202. Therefore, we repeated the experiment and now additionally use a specific monoclonal anti-PD-L1 antibody to block astrocyte-derived PD-L1 (Fig. 5m, Extended Data Fig. 1l).

9. The difference in Figure 5o between IFN and IFN+BMS202 is not significant for TNF and IL-6 upregulation. Also, the genes tested here are not the same genes as those in the “protective

signature” described at the beginning of the study. Why were different genes chosen for this part of the work?

We thank the Reviewer for this comment. We have repeated the experiment and expanded the genes representative for a pro-inflammatory activation of microglia (Fig. 5).

10. BMS202 is not introduced in the Methods. The only information given is that it is “small-molecule PD-L1 / PD-1 checkpoint inhibitor”. What is the formulation/preparation that was used for the intranasal delivery? The authors should add a detailed description in the Methods.

We thank the Reviewer for pointing this out. We have now added the information in the methods section of the revised manuscript.

11. Figure 3c. This referee could not find any description in the manuscript text, or the legends stating from what tissue the immune cells were isolated. It appears that these cells came from brain tissue, but this must be explicitly noted.

We apologize for this oversight and have now added additional information in the main text and figure legends that these cells were isolated from CNS (brain and spinal cord) of BMS202/vehicle treated mice.

Reviewer #2 (Remarks to the Author):

In this manuscript, the authors report on the regulation of chronic inflammatory processes by astrocytes via the immune checkpoint PD-L1/PD-1. This mechanism is investigated through the elegant combination of genetic perturbation studies and pharmacological approaches, both in vivo and in vitro. The authors show that PD-L1 expression is driven by aryl hydrocarbon receptor (AhR) and interferon signaling, and they ultimately propose the glial PD-L1/PD-1 axis as novel therapeutic target for the modulation of CNS intrinsic mechanisms relevant to progressive MS. The study is novel and relevant, yet there are several points the authors need to address:

1. Fig. 1: The conclusion that PDL1+ astrocytes are reparative based on the fact that they show lower expression of Ccl5 and Nos is an over statement. In fact, the gene profile shown in Fig 2 is not sufficient to support the conclusion that this population has a suppressed inflammatory phenotype. These conclusions should be toned town.

We thank the Reviewer for this helpful comment and agree that this claim cannot be made based on the expression of two genes only. Therefore, we have repeated the experiment, included more genes associated with an pro- or anti-inflammatory phenotype^{5,6} (Fig. 1f, Extended Data Fig. 1d), and toned down the statement as suggested by the reviewer.

2. Fig. 1: it is essential that the authors show PDL1 expression also in the normal brain to be able to claim that increased astrocytic PDL1 correlates with disease. Human?

This is an important point raised by the Reviewer and we appreciate the opportunity to clarify this point. Indeed, we observed that astrocytic PD-L1 expression is dependent on the inflammatory activation of astrocytes and almost absent in naïve mouse brain and spinal cord tissue (Fig. 1a-c). To corroborate this in human tissue, we have added a representative image for normal appearing white matter (NAWM) as control (Extended Data Fig. 1b).

3. Fig. 1: the authors show that sPDL1 is up in the CSF. Because in mice the authors measure the membrane bound form of PDL1, which is elevated, and sPDL1 is not measured, to better support the claim of correlation between human and mouse data, it would be important to show that expression/activity of metalloproteinases responsible of PDL1 cleavage is upregulated at peak EAE.

We thank the Reviewer for this important suggestion and now show the expression of the metalloproteases Adam10, Adam17, Mmp9, and Mmp13 by astrocytes throughout the EAE course (Extended Data Fig. 1f).

4. Fig. 2: there are inconsistencies in the way the data are presented: why is PDL1 expression after stimulation with different cytokines represented as % population in one case and MFI in another? Both % population and MFI should be reported in all cases, as they address different points, cell frequency and expression levels.

We appreciate the opportunity to clarify this point and agree, that each presentation (count, % of parent, MFI) confers different meaning that should carefully be considered when making a statement. To clarify this, we have adjusted manuscript and figures to now only show % of parent in the main figures, and supporting MFI in the Extended Data Figure (here represented as histogram of concatenated samples) throughout the manuscript.

5. Fig. 2: the fact that multiple STAT1-binding sites were identified in the Cd274 promoter by JASPAR is not confirmation that Trichostatin A, a suppressor of type-I interferon signaling,

abolished the induction of PDL1 expression by astrocytes. It opens that possibility, which still remains to be directly proven. This needs to be addressed or toned down.

We agree with the Reviewer that this statement cannot be made and accordingly rephrased our observation in the main text. In addition, we now show direct binding of STAT1 to the Cd274 promoter region following IFN- β stimulation of primary mouse astrocytes by ChIP (Fig. 2f). Together, this new dataset demonstrates that the induction of PD-L1 is positively regulated by interferon signaling.

6. PDL1 expression is increased by TNF/IL1b but to a much lower extent than with IFNs, both I and II. Furthermore, it's specifically IFN γ , not TNF or IL1b that seems to be the inflammatory stimulus pertinent to the MS environment that increases PDL1. There is selectivity in PDL1 expression based on these data. The authors should elaborate better on this point.

This is an important point raised by the Reviewer. To address this, we extended the number of stimuli (Fig. 2a), collectively demonstrating that indeed IFN- γ is the most potent stimulus, followed by IFN- β , in inducing Cd274. While this is the first report showing the regulation of Cd274 by Interferons in astrocytes, these effects have been demonstrated in a number of cell types⁷⁻⁹. Importantly, IFN- γ has been shown to be secreted by effector T cells in the context of MS and EAE, and reports have suggested opposing beneficial and detrimental roles of the cytokine for disease progression. However, recent data demonstrates that IFN- γ produced by NK cells has the capacity to induce anti-inflammatory functions in astrocytes, in the context of EAE and MS¹⁰. In accordance to the Reviewers comment, we now discuss this point in detail in the revised version of the discussion.

7. Fig. 3A: the EAE experiment in Fig. 3A is not convincing. The EAE severity is minimal, especially in the vehicle control where the scores barely go above 1 (no locomotor phenotype) and essentially go down to 0 within 10 days. This is odd and seems to indicate a failure of EAE. In fact, EAE in the treated group seems in line with what is expected in an untreated scenario. Ultimately, it doesn't look like the treatment made it worse, but that the EAE induction failed or was suboptimal. The experiment should be repeated.

We thank the Reviewer highlighting this weak spot and have repeated the experiment, now showing the clinical effect of intranasal BMS202 treatment with higher disease severity and incidence (94%, new Fig. 3a).

8. The fact that T cells in the periphery do not change is not an indication that "... nasal BMS202 treatment primarily affects inflammatory processes in the CNS without major peripheral effects". Immune cells are trafficking more into the CNS, as shown by the authors (Fig. 3C), which may be due to increased T cell proliferation and activation in the periphery prior to entering the CNS. Also, other myeloid populations should be assessed as well as B cells.

We thank the Reviewer for making this important point. In the revised version of the manuscript, we now show that number, cytokine production, and proliferation of peripheral T cells is not impacted by intranasal BMS202 administration. Furthermore, we have added the quantification of monocytes in the CNS and spleen, showing no effect of intranasal BMS202 treatment on the number of peripheral monocytes (Extended Data Fig. 3b-e). Furthermore, we have toned down this statement in the revised version of the manuscript.

9. It is unclear what the significance of the Ly6C+ and Ly6C- myeloid population is, what proportion of the peripheral myeloid population these are, and why they are specifically singled out.

We thank the Reviewer for the opportunity to clarify this point. We here refer to CD45^{hi}CD11b⁺Ly6C⁺ cells as inflammatory monocytes, while CD45^{hi}CD11b⁺Ly6C⁻ cells can include other myeloid cell populations like macrophages. Both compartments are a hallmark of the disease and can be found in MS lesions and EAE tissue, where they drive the inflammatory micromilieu. To make this more clear, we now show a gating strategy for the analysis used in Figure 3a-c (Extended Data Fig. 3a).

10. In several occasions, results are overinterpreted and conclusions overstated. For example, in results line 220: “Bulk RNA-seq of sorted microglia and astrocytes revealed increased pathogenic activity...”: expression changes of a few genes from a bulk sequencing experiment cannot be taken as indication of “increased pathogenic activity”. These statements need to be toned down.

We agree with the Reviewer that the functional relevance and pathogenicity of RNA-Seq data can only be inferred and apologize for this overstatement. While the analysis of transcriptional changes by gene-set enrichment analysis (GSEA) (Fig. 4i-j, Extended Data Fig. 5a-c) is commonly used to infer a pro- or anti-inflammatory polarization of cells, we have toned down this statement now and only describe the regulation of genes associated to pro- or anti-inflammatory functions.

11. Please check for typos and incorrect wording throughout. For example, legend Fig. 1d: the MFI diagrams are not histograms, please correct.

We thank the Reviewer for pointing this out. We have adjusted the text and corrected typos and incorrect wording throughout the manuscript.

Reviewer #3 (Remarks to the Author):

Key results

This article aims at investigating the immune regulation and tissue-protective functions of reactive astrocytes in the context of autoimmune CNS inflammation, by focusing of the PD-L1/PD-1 signaling. Authors show that PD-L1 is upregulated in astrocytes in the acute phase of EAE in mice and in acute inflammatory lesions on MS patient brain sections. Authors showed that PD-L1 is a downstream effector of both IFN β and AhR signaling. Using cell type specific approaches to knockout PD-L1 gene in astrocytes and PD-1 gene in microglia, authors showed a worsening of EAE phenotype and increased inflammatory mechanisms. Pharmacological blockade of PD-L1/PD-1 signaling worsens disease progression, suggesting that this signaling could be protective in both acute and progressive phases of EAE. A last sets of in vitro experiments suggest that PD-L1 can be secreted from astrocytes to bind PD1 and decrease the expression of pro-inflammatory cytokines in microglia.

Validity

1. Most in vivo manipulations of the PD-L1/PD-1 signaling do not use cell-type specific approaches (only experiments with lentiviral vectors) and because the astrocyte-specific upregulation of PD-L1 is not convincingly showed (see first § in the Significance section), it is possible that in EAE, activated microglia (and possibly other cell types) also upregulate PD-L1 and that PD-1 acts as an endocrine ligand. More importantly, there is no validation whatsoever that the CRISPR-Cas9-mediated knockout of PD-L1 gene in astrocytes and PD-1 gene in microglia after icv injections of lentiviral vectors approach is efficient and specific.

We thank the Reviewer for highlighting this important point and agree that it is important to show efficacy and specificity of various knockout approaches. We furthermore do not suggest that lentiviral delivery of CRISPR/Cas9 vectors is superior to transgenic models. Yet, it offers significant advantages in terms of time and resources. Several groups including ours have used lentiviral targeting of astrocytes for CRISPR/Cas9 mediated knockout¹⁰⁻¹², and while the efficacy is limited compared to transgenic approaches, the method offers high specificity. However, despite carrying a GFP tag, the reporter signal is hard to quantify in vivo (while we get good signal in vitro); we thus now directly quantify the expression of the targeted genes throughout the manuscript (as show in Extended Data Fig. 3g, Extended Data Fig. 4g, Extended Data Fig. 5i).

2. The timing of PD-L1 expression in astrocytes and of PD-1 in microglia over the course of EAE does not appear consistent. The number of PD-1+ microglia is maximal at recovery after EAE (Fig 5b) whereas PD-L1+ astrocytes levels are comparable to naïve mice at that time point (Fig 1c). Thus, it is not clear how this result can be reconciled with the author's conclusion that PD-L1 signaling in astrocytes also influence the progressive phases in EAE.

This is an important point raised by the Reviewer. To address this, we analyzed additional astrocytes from EAE brain and spinal cords, revealing a high expression of PD-L1 also in recovery stages (Fig. 1c). Moreover, we quantified astrocytic PD-L1 expression in combination with microglial PD-1 expression in an independent timecourse EAE experiment by flow cytometry (Extended Data Fig. 5f-g), which we now show and discuss in the revised version of the manuscript.

3. Some shortcuts are made between the interpretation of in vitro and in vivo experiments (line 56-57). For example, line 92: "concomitant" with should be replace by "these results are consistent with..." as it is just a parallel.

We thank the Reviewer for highlighting this and rephrased the respective sections accordingly.

4. There are instances where result interpretations were not supported by significant data. -Line 148, "IFN β and I3S significantly increased the expression of Cd274 by brain and spinal cord astrocytes": not true in spinal cord astrocytes, especially, because one outlier increases the mean Cd274 levels in the IFN β condition

We agree that this is an exaggerated statement. Therefore, we have generated and analyzed additional samples, showing significant upregulation of Cd274 by brain and spinal cord astrocytes derived from animals treated with IFN- β , while treatment with I3S only resulted in significant upregulation of Cd274 in spinal cord astrocytes (Fig. 2o, Extended Data Fig. 2m).

5. Line 254 "expression of pro-inflammatory cytokines": only IFN γ is significantly different, not IL-17 and GM-CSF

Thank you for pointing this out. We have corrected the statement, now referring to IFN- γ upregulation only.

6. Figure 5m The PD1 blocking antibody does not lead to significant decrease in PD1 expression

We thank the Reviewer for the opportunity to clarify this point. Indeed, antibody-mediated blockade of PD-1 does not necessarily lead to a downregulation, yet to a loss of its downstream functions. In these lines, we demonstrate an effect of PD-1 blockade on downstream inflammatory functions in microglia, now strengthened by the quantification of additional markers on protein level (Fig. 5k). Together, these data highlight the functional relevance of PD-1 controlled mechanisms and their antibody-mediated blockade.

7. Line 269 "suggesting that astrocytic PD-L1 controls pathogenic activities in microglia" this conclusion would have been true if authors used PD-L1 antibody instead of PD-1 antibodies

We agree with the Reviewer. In the revised version of the manuscript, we have added additional experiments demonstrating similar effects using anti-PD-L1 blockade (Fig. 5l-o, Extended Data Fig. 5l-m). Furthermore, we show anti-inflammatory effects following treatment with recombinant PD-L1, altogether suggesting that PD-L1 polarizes microglia towards protective phenotypes.

Significance

The main novelty is the attempt at dissecting astrocyte PD-L1 and microglia PD-1 crosstalk, but the in vivo experiments either using cell type isolation via flow cytometry and through genetic loss of function approaches are not fully convincing. In my opinion, authors included a significant amount of experiments using non cell type specific approaches (IFN β treatment etc) both in vivo and in vitro, which replicate the team's previous findings, to the expense of additional experiments to convincingly demonstrate that astrocytes play a key role in the protective effects of PD-L1/PD-1 signaling in EAE. Experiments on progressive stages of EAE are interesting.

We thank the reviewer for this assessment. Yet, we respectfully disagree with the notion of limited significance and mere replication of our previous results. In contrast, we strongly believe that this dataset adds important novel aspects relevant in the field of neuroimmunology, where glial PD-L1 / PD-1 interactions as well as their inducing cues (e.g. AhR) have not been shown before. Therapeutic targeting of this axis may furthermore pave the way to a novel avenue of treatment strategies in Multiple Sclerosis, as has been proposed for other neuropathologies¹³.

Most importantly, we thank the reviewer for highlighting the relevance of our results especially in progressive disease stages. Indeed, as of to date treatment options for progressive MS are limited and novel routes of treatment based on solid basic research data of high importance. In this light, our dataset adds novel aspects to progressive disease pathology and might thus contribute to overcome these limitations using innovative therapeutic approaches.

8. Detection of PD-L1 in astrocytes in acute inflammatory lesions on brain sections from MS patients has been described previously although the reference is not mentioned by authors (Pittet et al. The majority of infiltrating CD8 T lymphocytes in multiple sclerosis lesions is insensitive to enhanced PD-L1 levels on CNS cells. *Glia*. 2011 May;59(5):841-56)

We thank the Reviewer for pointing out this valuable reference. We now discuss the findings of Pittet et al. in the discussion of the revised manuscript. Moreover, we would like to point out that our manuscript adds novel data on the functional relevance and therapeutic addressability of glial PD-1 / PD-L1 signaling beyond the previously reported observations made in the references brought forward by the Reviewer. To reconcile, we have included this reference in the revised version of the manuscript.

Data and methodology

Some controls experiments are missing

Examples:

9. Validation of cell type specific isolation using flow cytometry (Figure 1b-c). Is there a negative selection for neurons?

In response to the Reviewers comment, we have analyzed additional astrocyte-specific markers to validate our sorting strategy (Extended Data Fig. 1c), which has been widely used and published before^{11,14-16}. While the isolation protocol itself has been optimized for CNS-resident populations, the retrieval of neurons remains a challenge to the field due to the disruption of their axons and processes during the isolation procedure. Accordingly, with the current protocol, we, and others only retrieve very few neuronal cells, as becomes apparent by using scRNA-Seq (Fig. 1b in Wheeler et al.¹¹), or flow cytometry (see below).

Figure 1. representative flow cytometric analysis of CNS tissue to demonstrate the low abundance of neurons (NeuN/CD171) using the digestion protocol described in the methods section.

10. Figure 1c: What about PD-L1 expression level in isolated microglia? This is an important control as there are reports that PD-L1 levels can also be expressed in microglia

This is true, and we also observe PD-L1 expression by microglia (Extended Data Fig. 1a-b, Extended Data Fig. 5h). In the revised version of the manuscript, we now show this both in MS and EAE tissue and discuss it in the main text. However, as demonstrated in Fig. 1d and Extended Data Fig. 1b, microglia take up a smaller fraction of PD-L1⁺ cells in the CNS compared to astrocytes. This also becomes clear in the publication by Pittet et al. ¹⁷, mentioned by the Reviewer, where the authors observe lower numbers of Iba1⁺PD-L1⁺ cells compared to GFAP⁺PD-L1⁺ cells. The exact role of PD-L1⁺ microglia will become subject to further investigation but exceeds the scope of this study.

11. Immunofluorescent stainings quality is poor: negative control (no primary antibody) for PD-L1? It is a bit surprising that a membrane-bound protein perfectly co-localizes with filamentous GFAP.

This is an important point raised by the Reviewer. Indeed, for all immunohistochemical analyses, negative controls have been used. We have added representative examples below.

Figure 2. representative images where the primary antibody for PD-L1 was omitted

Please also note that tissue shrinkage during freezing and fixation of the tissue affects co-localization of proteins. Moreover, intermediate filaments make contact with the cellular membrane, making it difficult to distinguish molecules that are exclusively membrane bound and intermediate filaments like GFAP (see below). Nevertheless, we included a representative image of an astrocyte demonstrating varying degrees of PD-L1 and GFAP overlap, with areas of low overlap (arrowheads) to areas with high overlap (orange) (scale bar 10µm).

Figure 3. representative images of varying degrees of PD-L1 / GFAP overlap in active white matter lesions in MS. Scale bar 30µm / scale bar 10µm (cropout).

12. There are few instances of where tool validation is limited or absent. No validation of successful targeting of astrocytes and microglia using icv injection of lentiviral vectors is missing, despite the construct carrying eGFP, which could allow identifying transduced cells.

As discussed in response to a prior comment, we agree that it is important to validate the efficacy of knockout approaches. As described before, despite carrying a GFP tag,

the reporter signal is hard to quantify in vivo (while we get good signal in vitro; see below); for this reason, we directly quantify the expression of the targeted gene (as shown in Extended Data Fig. 3g, Extended Data Fig. 4g, Extended Data Fig. 5i).

Figure 4. transduction efficiency of primary mouse astrocytes with lentiviral particles using lentiviral spinfection with 2µg/ml or 4µg/ml polybrene quantified by flow cytometry.

13. Line 141: There is no mention of the reference paper using these reporters (ref 56). Authors should mention that these experiments were performed in vitro and not in primary cell cultures of astrocytes but in HEK293T cells.

In response to the Reviewers comment, we have added this information and apologize for the oversight.

There is a lack of clarity when explaining the rationale for some experiments. In addition, the precise experimental approach is often not clearly mentioned in the results text (method, type of samples etc).

Examples:

14. Line 67: how PDL1+ astrocytes were identified? Authors should mention in the main text that it was done by flow cytometry (as opposed to immunostainings)

In response to the Reviewers comment, we rephrased the section and now describe that the analysis was done by flow cytometry.

15. Legend of Supp Fig 1a n=10 PD-L1 (samples)? From how many mice?

We thank the Reviewer for the opportunity to clarify this. In this case, 10 mice have been used to sort one PD-L1⁺ and PD-L1⁻ astrocyte sample from each mouse respectively. To avoid confusion, we have rephrased the description in the figure legend. Please also note that we have repeated the experiment with an additional set of mice and markers analyzed.

16. Which CNS region is used (brain or spinal cord), would be useful to add more details (which brain region? Which spinal cord level?) or justify that EAE-mediated inflammation is widespread and does not show prominent regional heterogeneity (grey vs white matter for example)

We have added this information throughout the manuscript. Indeed, we mostly use either brain and spinal cord or entire CNS, acknowledging for regional specificity as well as the importance of this signaling pathway throughout the CNS.

17. How MS lesion was identified ? Co-staining with immune cell markers? Demyelination?

We thank the Reviewer for the opportunity to clarify this point. MS lesions were identified by demyelination and immune cell infiltration as previously described¹⁸. We included the additional information in the methods section of the revised manuscript.

18. Line 114-115: “mouse and human astrocytes” authors should mention from primary cell cultures

We thank the Reviewer for highlighting this point and added the information that these are primary cell cultures.

19. Line 118: the rationale is not clearly stated. Authors should be explicit “because IFN primarily signals through STAT1...”

We agree with the Reviewer and accordingly rephrased the section in the revised version of the manuscript. We also added additionally data demonstrating direct binding of STAT1 to binding sites in the Cd274 promoter region in astrocytes.

20. Fig2f: relevance? Statistical analysis? There are probably many other transcription factor-binding sites. This result might not be critical to show, rather move to supplemental and replace by plot in SuppFig2g showing cd174 upregulation.

We thank the Reviewer for this suggestion and accordingly moved Fig. 2f into the Extended Data (now Ext. Data Fig. 2f), and Extended Data Fig. 2g into the main Figure (now Fig. 2h).

21. Supp Fig 4 h: why use LPS? These results are not presented or interpreted.

We thank the Reviewer for pointing this out. To avoid confusion, we excluded the data from the manuscript and only focus on IFN- γ stimulation in the revised version of the manuscript.

Figure and plot size

22. Overall, the figures and plots are very small. Authors should consider removing p value numbers and replace by stars to show significance.

We thank the reviewer for this comment. Yet, please note that the figure and font sizes have been chosen according to Nature formatting guidelines. Please also note that Nature encourages to provide “[...] exact p-values for both significant and non-significant P values.”, which is why we have decided to follow up on these guidelines throughout the manuscript.

Analytical approach

22. Sampling size and composition is often not clear. For example, for cell culture experiments, it is stated that n=3/condition and that the experiments were repeated 3 times. Were the data averaged?

We thank the Reviewer for the opportunity to clarify this. We only report the number of repetitions for in vivo EAE experiments. Here, the data is not averaged if not stated otherwise. Aside from this, the n values used throughout the manuscript describe the number of individual values/biological replicates used to calculate statistics.

23. Some stats are missing Figure 2 “o” p value for the comparison IFN β (written INF β) and I3S groups.

We thank the Reviewer for pointing this out. In the revised version of the manuscript we analyze additional samples from IFN and I3S treated mice with the respective statistics.

Suggested improvements

24. A schematic/graphical abstract summarizing the main findings would be useful

As suggested by the Reviewer, we now added a graphical abstract summarizing the key findings of the manuscripts.

25. Figure 1c: Additional RT-qPCR for astrocyte markers would be required to validate the isolation by flow cytometry

As described in a previous response to the Reviewers comment, we have now included additional astrocyte markers to validate the isolation by flow cytometry (Fig. 1c).

26. Line 76: Explain how these specific genes were chosen? (Fig 1f, easier to read if in the order as in SuppFig1a). Did authors check the expression anti-inflammatory genes?

We thank the Reviewer for the opportunity to clarify this. Indeed, the markers analyzed include established pro-inflammatory (Ccl5, Nos2) and anti-inflammatory (Tgfb1, Lif, Ptn; Ngf, Gdnf) markers. Please note that we have significantly expanded the number of analyzed genes (Fig. 1f, Extended Data Fig. 1d) and now also include non-secreted markers of astrocyte activation (e.g. Nfkb, Cd44).

27. It would be interesting to determine whether PD-L1 is differentially upregulated in different types of lesions (as done in Schirmer et al. 2014)

This is an interesting point raised by the Reviewer. Indeed, in the study recommended by the Reviewer from Pittet et al. ¹⁷, the authors demonstrate higher numbers of PDL1⁺GFAP⁺ astrocytes in subacute lesion areas compared to acute.

27. On all clinical score plots: only half of the error bars (upwards on the top curve and downwards on the bottom curve) which is misleading. Both error bars should be shown in all plots.

In response to the Reviewers comment, we now show error bars in both directions for EAE graphs.

28. Gender considerations: Experiments in EAE mice were performed in females only but not justified (add reference to the literature).

EAE studies are commonly performed in female mice, as pointed out in original work ¹⁹, partly due to differences in susceptibility and disease severity ^{20,21}. Moreover, please also note that MS is up to 4 times more common in females, which is why we chose to not include male mice here. Indeed, this is a problem common to the field that needs to be addressed in future analyses and studies.

29. For histological analysis of human brain samples from controls and MS patients, gender is not provided

We thank the Reviewer for pointing this out and apologize for missing this information in the original version. The human samples analyzed by immunohistochemistry in this manuscript were obtained from a female MS patient, tissue that has previously been used in prior publications by our group ¹⁴. This information can also be found in the reporting summary and methods section of the revised manuscript.

Minor comments:

29. Line 172 and 174: Knockout rather than inactivation or knockdown

In the revised version of the manuscript, we now refer to knockout.

30. Line 268 add “in microglia”

We have extensively rewritten the respective section and included new data in the revised version of the manuscript.

31. Figure 2 d in legend replace “mouse” by “human” astrocytes

We thank the Reviewer for pointing this out. In the revised version of the manuscript, we have moved MFI histograms into the Extended Data Fig. 2 and adjusted the figure legend accordingly.

32. Figure 2 h not useful without gene names

In response to the Reviewers comment, we now show additional gene names in the revised version of the manuscript

33. Figure 2 m legend missing

We have not included a legend for the color coding, as all groups are described on the x-axis labels and apologize for the oversight.

34. Figure 2 k vehicle tag in black, not grey

We thank the Reviewer for pointing this out and changed the color to black.

35. Figure 2 o and n should be swapped to improve readability

In the revised version of the manuscript, we added additional data and restructured Figure 2, now showing Fig. 2 o and Fig. 2 n next to another.

36. §line 195 Rather belongs in the introduction, cut is short in the results

We thank the Reviewer for this suggestion and rephrased the beginning of this section accordingly. We now discuss this in more detail in the introduction (lines 50-56), and excluded it in the result section.

37. Line 370 GFA-ABC1D or gfa2 which one?

We thank the Reviewer for pointing this out. The promoter used is the ABC₁D GFAP promoter, which we now describe in the method section.

38. § line 505: missing to mention PDL1 antibody for FACS isolation of astrocytes?

We thank the Reviewer for highlighting this. Indeed, the antibody used for the isolation of PD-L1⁺ astrocytes is the same, we also used for general flow cytometry staining described in the paragraph above. We now also describe the use of the antibody in the subsequent section.

39. § line 519: primers or sequences for other genes used throughout the manuscript?

We thank the Reviewer for highlighting this and added the missing information for all genes analyzed in the manuscript.

40. Line 570: Normal appearing white matter abbreviated only

We thank the Reviewer for pointing this out and adjusted the text accordingly.

Clarity and context

Some words or concept are not introduced, described or explained

Examples:

41. Line 121 "ISGF3" not explained

Since the statement does not add to the interpretation of the results, we have rephrased the section and now exclude "ISGF3" in the revised version of the manuscript.

42. Line 135: GFAP-cre; AcHfl/fl mouse model description is missing from the material and method section. In addition, cd174 levels are not significantly decreased. The downregulation is mild but visually enhanced on the plot because the axis is cut

We thank the Reviewer for the opportunity to clarify this point. The publicly available data was obtained from Rothhammer et al. ¹⁴, where the mouse model is extensively described and characterized, as referenced in the respective section. In response to the Reviewers comment, we added statistics and show the plot without segmented y-axis.

43. Line 141: There is no mention of the reference paper using these reporters (ref 56). Authors should mention that these experiments were performed in vitro and not in primary cell cultures of astrocytes but in HEK293T cells.

In response to the Reviewers comment, we now mention the use of HEK293T cells not only in the methods section but also in the main text.

44. Line 190: The result interpretation concerning the recovery phase in the EAE mouse model is not straightforward

In response to the Reviewers comment, we rephrased the section to make our interpretation of the results more clear and thank the reviewer for his insightful comment.

1. Baruch K, Deczkowska A, Rosenzweig N, et al. PD-1 immune checkpoint blockade reduces pathology and improves memory in mouse models of Alzheimer's disease. *Nat Med*. 2016;22(2):135-137. doi:10.1038/nm.4022
2. Baruch K, Rosenzweig N, Kertser A, et al. Breaking immune tolerance by targeting Foxp3+ regulatory T cells mitigates Alzheimer's disease pathology. *Nat Commun*. 2015;6:7967. doi:10.1038/ncomms8967
3. Rosenzweig N, Dvir-Szternfeld R, Tsitsou-Kampeli A, et al. PD-1/PD-L1 checkpoint blockade harnesses monocyte-derived macrophages to combat cognitive impairment in a tauopathy mouse model. *Nat Commun*. 2019;10(1):465. doi:10.1038/s41467-019-08352-5
4. Marsh SE, Abud EM, Lakatos A, et al. The adaptive immune system restrains Alzheimer's disease pathogenesis by modulating microglial function. *Proc Natl Acad Sci U S A*. 2016;113(9):E1316-E1325. doi:10.1073/pnas.1525466113
5. Sofroniew MV. Astrocyte barriers to neurotoxic inflammation. *Nat Rev Neurosci*. 2015;16(5):249-263. doi:10.1038/nrn3898
6. Colombo E, Farina C. Astrocytes: Key Regulators of Neuroinflammation. *Trends in Immunology*. 2016;37(9):608-620. doi:10.1016/j.it.2016.06.006
7. Garcia-Diaz A, Shin DS, Moreno BH, et al. Interferon Receptor Signaling Pathways Regulating PD-L1 and PD-L2 Expression. *Cell Rep*. 2017;19(6):1189-1201. doi:10.1016/j.celrep.2017.04.031
8. Lee SJ, Jang BC, Lee SW, et al. Interferon regulatory factor-1 is prerequisite to the constitutive expression and IFN-gamma-induced upregulation of B7-H1 (CD274). *FEBS Lett*. 2006;580(3):755-762. doi:10.1016/j.febslet.2005.12.093
9. Escors D, Gato-Cañas M, Zuazo M, et al. The intracellular signalosome of PD-L1 in cancer cells. *Signal Transduct Target Ther*. 2018;3:26. doi:10.1038/s41392-018-0022-9
10. Sanmarco LM, Wheeler MA, Gutiérrez-Vázquez C, et al. Gut-licensed IFN γ + NK cells drive LAMP1+TRAIL+ anti-inflammatory astrocytes. *Nature*. 2021;590(7846):473-479. doi:10.1038/s41586-020-03116-4
11. Wheeler MA, Clark IC, Tjon EC, et al. MAFG-driven astrocytes promote CNS inflammation. *Nature*. 2020;578(7796):593-599. doi:10.1038/s41586-020-1999-0
12. Linnerbauer M, Lößlein L, Farrenkopf D, et al. Astrocyte-Derived Pleiotrophin Mitigates Late-Stage Autoimmune CNS Inflammation. *Frontiers in Immunology*. 2022;12. Accessed July 12, 2022. <https://www.frontiersin.org/articles/10.3389/fimmu.2021.800128>
13. Schwartz M, Arad M, Ben-Yehuda H. Potential immunotherapy for Alzheimer disease and age-related dementia. *Dialogues Clin Neurosci*. 2019;21(1):21-25.
14. Rothhammer V, Borucki DM, Tjon EC, et al. Microglial control of astrocytes in response to microbial metabolites. *Nature*. 2018;557(7707):724-728. doi:10.1038/s41586-018-0119-x
15. Rothhammer V, Maccanfroni ID, Bunse L, et al. Type I interferons and microbial metabolites of tryptophan modulate astrocyte activity and central nervous system inflammation via the aryl hydrocarbon receptor. *Nat Med*. 2016;22(6):586-597. doi:10.1038/nm.4106

16. Chao CC, Gutiérrez-Vázquez C, Rothhammer V, et al. Metabolic Control of Astrocyte Pathogenic Activity via cPLA2-MAVS. *Cell*. 2019;179(7):1483-1498.e22. doi:10.1016/j.cell.2019.11.016
17. Pittet CL, Newcombe J, Antel JP, Arbour N. The majority of infiltrating CD8 T lymphocytes in multiple sclerosis lesions is insensitive to enhanced PD-L1 levels on CNS cells. *Glia*. 2011;59(5):841-856. doi:10.1002/glia.21158
18. Alvarez JI, Saint-Laurent O, Godschalk A, et al. Focal disturbances in the blood-brain barrier are associated with formation of neuroinflammatory lesions. *Neurobiol Dis*. 2015;74:14-24. doi:10.1016/j.nbd.2014.09.016
19. Wekerle H, Kojima K, Lannes-Vieira J, Lassmann H, Linington C. Animal models. *Annals of Neurology*. 1994;36(S1):S47-S53. doi:10.1002/ana.410360714
20. Papenfuss TL, Rogers CJ, Gienapp I, et al. Sex differences in experimental autoimmune encephalomyelitis in multiple murine strains. *Journal of Neuroimmunology*. 2004;150(1):59-69. doi:10.1016/j.jneuroim.2004.01.018
21. Wiedrick J, Meza-Romero R, Gerstner G, et al. Sex differences in EAE reveal common and distinct cellular and molecular components. *Cell Immunol*. 2021;359:104242. doi:10.1016/j.cellimm.2020.104242

REVIEWERS' COMMENTS

Reviewer #1 (Remarks to the Author):

In the revised version of their manuscript entitled "PD-L1 positive astrocytes ameliorate neuroinflammation by interacting with PD-1 positive microglia", the authors claim that they addressed all the referees' comments, and incorporated all the requested textual changes. Yet, one of the key comments raised by the reviewers was that the authors were not sufficiently careful in distinguishing between the various types of brain inflammation associated with remitting relapsing autoimmune inflammation, and the low-grade innate inflammation seen in neurodegenerative diseases. This concern was addressed in the point-by-point response to Reviewer #1, but not comprehensively in the text, and certainly should be corrected in the title "PD-L1 positive astrocytes ameliorate AUTOIMMUNE neuroinflammation by interacting with PD-1 positive microglia. In addition, this issue should be emphasized in the Discussion, given the fact that even in the animal model, the PD-L1 positive astrocytes could display opposite effects during remission versus the relapse phase.

Reviewer #2 (Remarks to the Author):

Overall, I am satisfied with the changes made by the authors. However, one additional point should be addressed: the EAE graphs are often extremely small and difficult to read. One example is the graph in Fig. 3i, which is unintelligible with overlapping curves that blend into each other. Furthermore, the statistical analysis for the EAE curves should also be revisited. In Fig. 3i, the authors use an unpaired t-test when the appropriate test is a repeated measure ANOVA. This has to be done for all EAE graphs.

Reviewer #3 (Remarks to the Author):

Authors took into account all the reviewer's points, which significantly increased the quality of the manuscript, especially regarding additional control experiments.

I still have two comments:

-I find it surprising that by adding more astrocytes to the analysis of Fig.1c, the expression pattern of PD-L1 changes drastically, which now is fitting to the reviewer's remark. On Extended figure 5f-g, the expression of PD-L1 in spinal cord astrocytes seems to drop during the recovery phase, which is not observed in Figure 1c. Could this be explained by the slightly different time points that were chosen based on the clinical score between Fig 1 and Extended Fig 5? A statistical analysis should be added for the time course plots in Extended figure 5f-g.

-Authors response to the reviewer's remark on the manuscript significance is not convincing. They could have precisely identified which «important novel aspects relevant in the field of neuroimmunology» they are referring to

Minor comments:

- colored dots in legend are missing below plot in Extended Figure 5i
- Clinical score numbers (0 to 3) below each plots in Extended Fig 5g would make it easier to read
- Add marker name within each single channel image in Figure 1d
- Typo in Extended data figure 3d

Reviewer #1 (Remarks to the Author):

In the revised version of their manuscript entitled "PD-L1 positive astrocytes ameliorate neuroinflammation by interacting with PD-1 positive microglia", the authors claim that they addressed all the referees' comments, and incorporated all the requested textual changes. Yet, one of the key comments raised by the reviewers was that the authors were not sufficiently careful in distinguishing between the various types of brain inflammation associated with remitting relapsing autoimmune inflammation, and the low-grade innate inflammation seen in neurodegenerative diseases. This concern was addressed in the point-by-point response to Reviewer #1, but not comprehensively in the text, and certainly should be corrected in the title "PD-L1 positive astrocytes ameliorate AUTOIMMUNE neuroinflammation by interacting with PD-1 positive microglia. In addition, this issue should be emphasized in the Discussion, given the fact that even in the animal model, the PD-L1 positive astrocytes could display opposite effects during remission versus the relapse phase.

We thank the reviewer for his remark and apologize for this inconsistency. To consequently address this point, we have adjusted the manuscript throughout and changed the respective paragraphs to strictly refer to autoimmune inflammation only. In these lines, we have also changed the title of our manuscript to refer to autoimmune neuroinflammation explicitly. In the revised discussion section of our manuscript, we now highlight the need of further investigations to delineate whether the observed mechanisms are limited to autoimmune neuroinflammation only or also operational in the context of low-grade innate inflammation seen in neurodegenerative conditions. With these alterations, we believe to now clearly make the point that our observations refer to autoimmune inflammation at this stage only and have clarified this point throughout the manuscript as well as in the revised version of the discussion.

Reviewer #2 (Remarks to the Author):

Overall, I am satisfied with the changes made by the authors.

We thank the Reviewer for his positive assessment of the changes made to the manuscript.

However, one additional point should be addressed: the EAE graphs are often extremely small and difficult to read. One example is the graph in Fig. 3i, which is unintelligible with overlapping curves that blend into each other. Furthermore, the statistical analysis for the EAE curves should also be revisited. In Fig. 3i, the authors use an unpaired t-test when the appropriate test is a repeated measure ANOVA. This has to be done for all EAE graphs.

In response to the Reviewers comment, we have increased the size of Fig. 3i. We would furthermore like to point out that in Figure 3i, the area under the curve was calculated, as has been performed in PMID: 33414215. Statistical testing was performed using Tukey's multiple comparisons test. The exact statistical tests are pointed out in the individual figure legends.

Reviewer #3 (Remarks to the Author):

Authors took into account all the reviewer's points, which significantly increased the quality of the manuscript, especially regarding additional control experiments.

We would like to thank the reviewer for his positive assessment of the revisions made to our manuscript.

I still have two comments:

-I find it surprising that by adding more astrocytes to the analysis of Fig.1c, the expression pattern of PD-L1 changes drastically, which now is fitting to the reviewer's remark. On Extended figure 5f-g, the expression of PD-L1 in spinal cord astrocytes seems to drop during the recovery phase, which is not observed in Figure 1c. Could this be explained by the slightly different time points that were chosen based on the clinical score between Fig 1 and Extended Fig 5? A statistical analysis should be added for the time course plots in Extended figure 5f-g.

We agree with the Reviewer that the difference in the expression of PD-L1 by spinal cords astrocytes in Figure 1 and Supplementary Figure 4g are likely due to differences in the timepoints analyzed. While in Figure 1c, astrocytes were analyzed during the recovery stage of EAE, Supplementary Figure 5g depicts their expression during a period of clinical worsening. While further studies will be needed to support this claim, one could speculate that the decreased expression of PD-L1 and the associated loss of its immune-regulatory functions during these stages may perpetuate the worsening of the disease.

-Authors response to the reviewer's remark on the manuscript significance is not convincing. They could have precisely identified which «important novel aspects relevant in the field of neuroimmunology» they are referring to

We thank the reviewer for his remark. In order to furthermore underline the significance of our manuscript, we would like to point out that the control of PD-1⁺ microglia by PD-L1⁺ astrocytes in the context of autoimmune CNS inflammation has not been proposed before, which constitutes the major finding of our manuscript. In response to the reviewer's comment, we have included this and other aspects in the revised version of the discussion.

Minor comments:

-colored dots in legend are missing below plot in Extended Figure 5i

-Clinical score numbers (0 to 3) below each plots in Extended Fig 5g would make it easier to read

-Add marker name within each single channel image in Figure 1d

-Typo in Extended data figure 3d

We thank the Reviewer for pointing this out. We have made the respective corrections in the revised version of the manuscript.